# Cryo-EM of mammalian PA28αβ-iCP immunoproteasome reveals a distinct mechanism of proteasome activation by PA28αβ

Jinhuan Chen[1,5], Yifan Wang[1,2,5], Cong Xu[1,2,5], Kaijian Chen[1,2], Qiaoyu Zhao[1,2], Shutian Wang[1,2], Yue Yin[3], Chao Peng ◉ [3✉], Zhanyu Ding[1✉] & Yao Cong ◉ [1,2,4✉]

The proteasome activator PA28αβ affects MHC class I antigen presentation by associating with immunoproteasome core particles (iCPs). However, due to the lack of a mammalian PA28αβ-iCP structure, how PA28αβ regulates proteasome remains elusive. Here we present the complete architectures of the mammalian PA28αβ-iCP immunoproteasome and free iCP at near atomic-resolution by cryo-EM, and determine the spatial arrangement between PA28αβ and iCP through XL-MS. Our structures reveal a slight leaning of PA28αβ towards the α3-α4 side of iCP, disturbing the allosteric network of the gatekeeper α2/3/4 subunits, resulting in a partial open iCP gate. We find that the binding and activation mechanism of iCP by PA28αβ is distinct from those of constitutive CP by the homoheptameric *Tb*PA26 or *Pf*PA28. Our study sheds lights on the mechanism of enzymatic activity stimulation of immunoproteasome and suggests that PA28αβ-iCP has experienced profound remodeling during evolution to achieve its current level of function in immune response.

[1] State Key Laboratory of Molecular Biology, National Center for Protein Science Shanghai, Shanghai Institute of Biochemistry and Cell Biology, Center for Excellence in Molecular Cell Science, Chinese Academy of Sciences, Shanghai 200031, China. [2] University of Chinese Academy of Sciences, Beijing 100049, China. [3] National Facility for Protein Science in Shanghai, Zhangjiang Lab, Shanghai Advanced Research Institute, CAS, Shanghai 201210, China. [4] Shanghai Science Research Center, CAS, Shanghai 201210, China. [5]These authors contributed equally: Jinhuan Chen, Yifan Wang, Cong Xu. ✉email: pengchao@sari.ac.cn; dingzhanyu@sibcb.ac.cn; cong@sibcb.ac.cn

Proteasomes degrade many protein substrates in the cytosol and nuclei of eukaryotic cells, and are hence essential for many aspects of cellular function[1–3]. Proteasome can refer to a variety of complexes whose enzymatic cores are the cylindrical 20S proteasome (also termed core particle, CP). Proteasome activity is fine-tuned as a result of the association of the proteolytic CP with diverse proteasomal activators (PAs), including PA700, PA28, and PA200, that can induce the opening of the CP gate into the central proteolytic cavity, allowing substrates to be degraded[3]. PA700 (also called 19S), an ATP-dependent activator, could bind a CP to form the 26S proteasome, which mediates degradation of ubiquitylated substrates[4,5]. In contrast, PA28 (also called 11S or REG in most organisms, and PA26 in *Trypanosoma brucei*, *Tb*PA26) stimulates the degradation of peptides in an ATP/ubiquitin-independent manner[6,7]. In the PA28 family, the homologous PA28α and PA28β, whose expressions are induced by IFN-γ, can associate into a heteroheptamer[8–12], while PA28γ can form a homoheptamer[13]. PA200, also called Blm10 in yeast, is another ATP/ubiquitin-independent proteasome activator[14–17].

PA28αβ is usually linked to major histocompatibility complex (MHC) class I antigen processing, a critical step in immune response[18,19]. PA28αβ has been shown in vitro to affect the generation of peptides by proteasome CPs and is required for efficient presentation of many T cell epitopes from a number of viral, bacterial, and tumor-derived antigens[18]. PA28 deficiency could reduce the production of MHC class I-binding peptides in cells[20–22].

Most tissues and cells express predominantly the constitutive 20S proteasome (cCP) with the proteolytic active sites located at the β1, β2, and β5 subunits of the CP[23–25]. However, lymphoid cells and cells exposed to cytokines such as IFN-γ alternatively express three homologous subunits (β1i/LMP2, β2i/MECL-1, and β5i/LMP7), replacing the constitutive ones, in the 20S immunoproteasome (iCP) particles—with this alternative expression resulting in a change in the proteolytic activities[26–31]. It has been suggested that iCPs generate class I-binding peptides to participate in antigen processing and play an important role in MHC class I antigen presentation[32–34].

With constant efforts during the past years, the structural bases of the 19S-cCP and Blm10-cCP systems have become better understood[3,35–39]. Recently, a crystal structure of mouse PA28$_4$β$_3$ revealed an alternating arrangement of four α and three β chains[40]. Still, structural studies on the PA28-CP complex are mostly limited to the homoheptameric *Tb*PA26 or *Plasmodium falciparum* (*Pf*) PA28 in complex with the cCP[41–43]. A complete structure of a mammalian PA28αβ-cCP or PA28αβ-iCP proteasome has not yet been determined. This deficiency is mostly due to the lability and sensitivity to salt, the binding of mammalian PA28αβ to the CP is reversible[44]. These issues make the in vitro assembly of the PA28αβ-CP complex or direct isolation of the complex from tissues or cells extremely challenging[45]. As a result, the mechanisms by which the heteroheptameric PA28αβ binds and activates the CP or iCP remain elusive.

Here, we assemble a mammalian PA28αβ-iCP immunoproteasome complex from human PA28αβ and bovine spleen iCP. We determine the uncharacterized cryo-EM structures of free bovine iCP and the single- and double-capped PA28αβ-iCP and PA28αβ-iCP-PA28αβ to resolutions of 3.3, 4.1, and 4.2 Å, respectively. We also depict the spatial arrangement between PA28αβ and iCP by chemical cross-linking coupled mass spectrometry (XL-MS) analysis. Importantly, our study reveals a distinct mechanism for the binding and activation of iCP by the heteroheptameric PA28αβ, compared with those of cCP by the homoheptameric *Tb*PA26 or *Pf*PA28. We also find conserved

differences between the immune catalytic subunits and those of the constitutive ones, beneficial for the development of immune-specific inhibitors. Our study provides insights into the unique mechanism of proteasome activation induced by PA28αβ, and how this non-ATPase activator regulates CP gate opening potentially through an on-and-off mode for substrate processing.

## Results

**Cryo-EM structures of mammalian PA28αβ-iCP immunoproteasomes.** To avoid the known difficulties in isolating an intact and stable PA28αβ-iCP complex directly from tissues or cells for cryo-EM study[45], we first expressed and purified human PA28αβ heteroheptamer from *Escherichia coli* following the established procedure (Supplementary Fig. 1a)[9,40,46], and also isolated bovine 20S proteasome directly from bovine spleen (Supplementary Fig. 1b). Note that we selected the spleen because it has been reported that immunoproteasome is the most important proteasome subtype (more than 70% of the total 20S proteasome pool) in spleen[47–49]. Our MS analysis also showed that the abundance of immune-β subunits, indicated by peptide-spectrum-matches value, is obviously higher (>70%) than that of standard-β subunits (Supplementary Table 1), suggesting that iCP is the main proteasome CP purified from bovine spleen, while cCP is rather less populated. We then in vitro assembled the purified human PA28αβ and bovine iCP into the intact PA28αβ-iCP immunoproteasome complex in the presence of glutaraldehyde as a cross-linker (Supplementary Fig. 1c, d). Cross-linking has become a more adopted strategy in cryo-EM studies to stabilize fragile micromolecular complexes[42,50–53]. Our further in vitro proteolytic activity assay against the fluorogenic peptide Suc-LLVY-AMC showed that the reconstituted PA28αβ-iCP complex was functionally active (Supplementary Fig. 1e).

From the same set of cryo-EM data of the mammalian PA28αβ-iCP complex, we resolved three maps, including the free bovine iCP, the single-capped PA28αβ-iCP and the double-capped PA28αβ-iCP-PA28αβ at the resolutions of 3.3, 4.1, and 4.2 Å, respectively (Fig. 1a–d, Supplementary Fig. 2, and Supplementary Table 2). To the best of our knowledge, none of these structures has been determined previously. Our PA28αβ-iCP and PA28αβ-iCP-PA28αβ maps revealed a funnel-like PA28αβ heteroheptamer associated with, respectively, one or both ends of the iCP (Fig. 1a, b). PA28αβ was observed to be ~90 Å in diameter and ~90 Å in height, and to consist of a central channel of 35 Å in diameter at the 20S-binding end and 20 Å at the distal end (Fig. 1a). The size of the channel is comparable to those of the homologous PA26 and PA28[40,42]. Note that our cryo-EM maps revealed extra pieces of density extending on top of the PA28αβ core (Fig. 1a, b), to some extent covering the entrance to the funnel; these pieces of density were most likely derived from the dynamic apical loops. Additional local resolution analysis suggested that in both capped complexes, PA28αβ appeared less well resolved than did the complexed iCP, indicating the intrinsic dynamic nature of PA28αβ, especially in its unstructured apical loop region (Fig. 1e, f). The dynamic nature of the apical loops may potentially be beneficial for substrate recruitment and substrate entry into or efflux out of the central channel of PA28αβ.

For the free iCP, dominant portion of the map showed local resolution levels better than 3.0 Å (Fig. 1g), revealing sidechain densities for most of the amino acids. Note that the unambiguous assignment of amino-acid sidechains here enabled us to confirm that the isolated bovine spleen 20S proteasome was indeed mostly immunoproteasome, for instance, for these distinct regions between iCP and cCP, overall iCP model fits into our cryo-EM

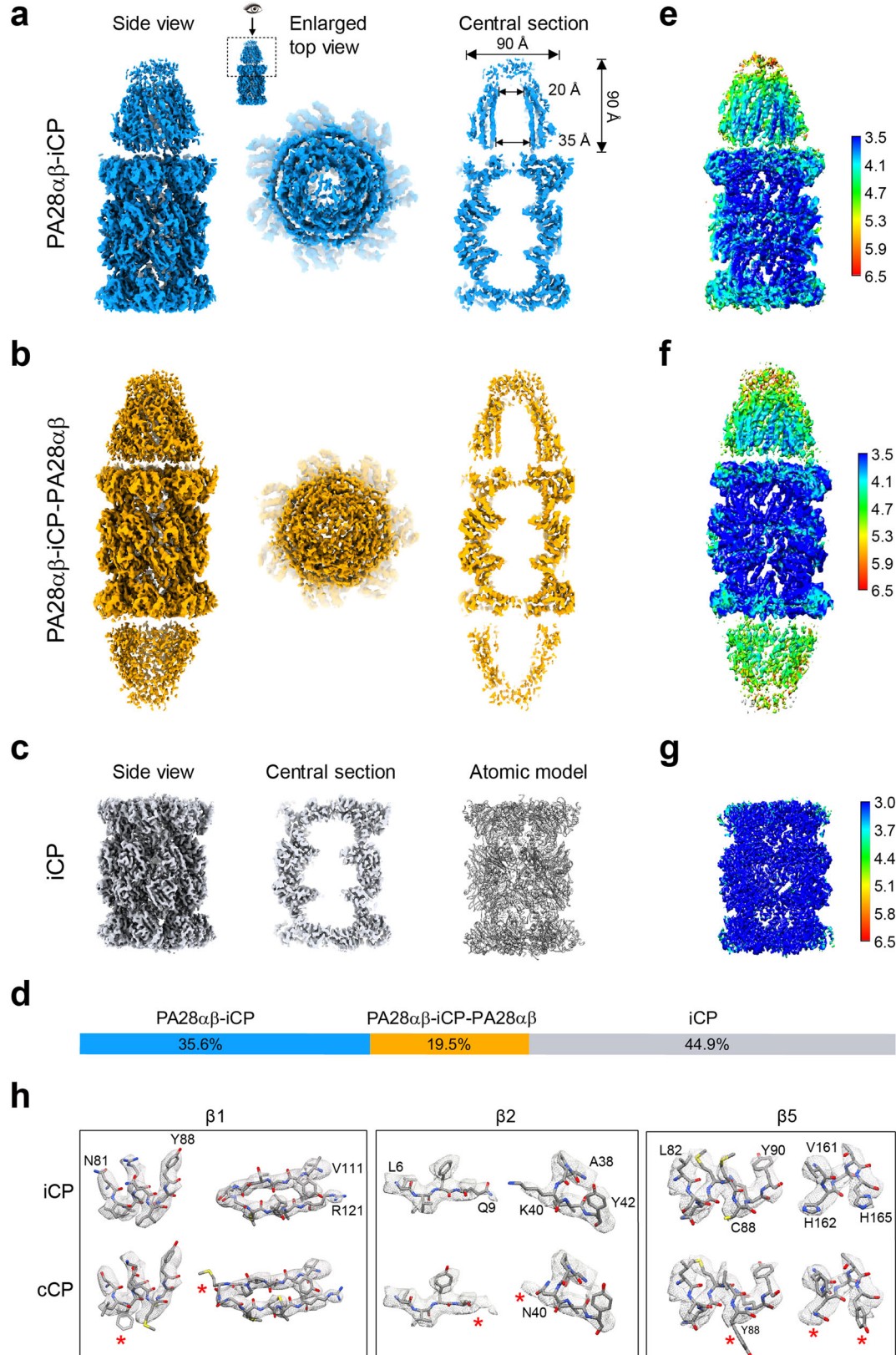

map better (Fig. 1h), only rare locations imply the partial contribution from the signal of cCP in addition to the dominant signal from iCP (Supplemental Fig. 2d). This further implied that the dominantly populated iCP particles primarily contributed to our cryo-EM maps; still, we cannot exclude the minor contribution from the signal of cCP.

**Relative spatial arrangement between PA28αβ and iCP determined by XL-MS.** A recent crystal structure of mouse PA28α₄β₃ revealed an alternating arrangement of four α and three β chains with two consecutive α subunits sitting side-by-side[40]. Due to the high sequence identity (~94%) between human and mouse PA28αβ (Supplementary Fig. 3) and the similar expression and

**Fig. 1 Overall cryo-EM maps of mammalian PA28αβ-iCP, PA28αβ-iCP-PA28αβ, and iCP. a** Cryo-EM map of the single-capped PA28αβ-iCP complex (side view, enlarged top view, and one central section). The inset illustrates the visualization angle and region, which was followed in the top view generation of PA28αβ-iCP-PA28αβ. **b** Cryo-EM map of the double-capped PA28αβ-iCP-PA28αβ complex, in the same rendering style as in (**a**). **c** Cryo-EM structure of the free iCP reconstructed from the same dataset (side view, one central section, and atomic model). **d** The relative populations of PA28αβ-iCP (dodger blue), PA28αβ-iCP-PA28αβ (gold), and iCP (gray). **e-f** Local resolutions of the PA28αβ-iCP (**e**), PA28αβ-iCP-PA28αβ (**f**), and iCP (**g**) cryo-EM maps determined using ResMap. Shown are the cut-away view of the corresponding density map, with the color bar on the right labeling the resolutions (in Å). **h** Identifications of immunoproteasome amino-acid residues by fitting details for the free iCP. The iCP sequence (top row), but not the conventional CP sequence (bottom row), fits well into the density of our free iCP cryo-EM map in the three catalytic subunits. The red asterisks indicate the residues of cCP that do not match well with the density.

purification procedure undertaken by us and in their study[40], the subunit ordering of human PA28αβ heteroheptamer was expected to be the same to that of mouse PA28αβ[40], which is also the case for the subunit ordering of the more complexed eukaryotic chaperonin TRiC/CCT consisting of eight paralogous subunits[54–57]. To further delineate the relative spatial arrangement between the PA28αβ and CP units, we carried out an XL-MS analysis on the PA28αβ-iCP complex. The cross-links detected within the CP of proteasome fulfill the spatial geometry constrains of the linked amino acids, validating the reliability of our XL-MS data (Supplementary Fig. 4). We then focused on the cross-linked contacts identified in the interface between the PA28αβ and iCP units to exclude the intra-units interactions. This analysis disclosed a number of interactions involving the C termini of the PA28α or PA28β subunit and the amino-acid residues lying in the iCP α-ring pocket regions, including PA28α-α2, PA28α-α3, PA28β-α2, and PA28β-α6 interactions (Fig. 2a and Supplementary Table 3). These interaction constraints led to the deciphering of a unique spatial arrangement of the PA28αβ and iCP units relative to each other, with the two consecutive PA28α subunits ($\alpha_1$ and $\alpha_4$) residing on top of the α6 and α7 subunits of iCP (Fig. 2b).

Based on this relative spatial arrangement of the PA28αβ and iCP units, we built a pseudo-atomic model for each of the PA28αβ-capped complexes (Fig. 2c, d). Interestingly, our structure revealed a slight tilt (~3°) of PA28αβ toward the α3–α4 side of the iCP (Fig. 2c), comparable to that observed for the PfPA28-CP complex[42]. This lack of alignment of the axes of the PA28αβ and iCP units was reminiscent of a resting state of 26S proteasome with the Rpt ring lean sitting on the plane of 20S CP[39].

**A unique mode of interaction between the mammalian PA28αβ activator and iCP.** Regarding the interaction interface between PA28αβ and iCP, our PA28αβ-iCP map revealed relatively strong pieces of density corresponding to four PA28αβ C-terminal tail insertions into the α-ring pockets of iCP, including the C termini of PA28 $\beta_3$, $\alpha_3$, $\beta_2$, and $\alpha_2$ inserted into, respectively, the pockets of α1/2, α2/3, α3/4, and α4/5 of iCP, and a rather weak piece of density for the C terminus of PA28$\beta_1$ in the α5/6 pocket (Fig. 3a, b). However, we found no obvious extra density in the pockets of α6/7 and α7/1 (Fig. 3a, b). Our additional binding analysis and proteolytic activity assay showed that after C-terminal truncations (truncated K245-Y249 for PA28α and E234-Y239 for PA28β, together termed PA28αβ$^{\Delta C\text{-}tails}$), PA28αβ$^{\Delta C\text{-}tails}$ cannot bind iCP (Supplementary Fig. 5a, b), consequently, it failed to activate iCP (Supplementary Fig. 5d). These biochemical data substantiate the notion that the PA28αβ C-terminal tail insertions into the iCP pockets can stabilize the binding of activator but cannot open the iCP gate, in line with previous reports of the homologous PA26-CP systems[41,43,58,59].

Interestingly, PA28αβ overall displayed stronger interactions with the α3–α4 side of iCP but no binding with the opposite α6–

α7 side (Fig. 3a), consistent with our observations of the slight leaning of PA28αβ towards the α3–α4 side and more intimate interactions with iCP on this side (Fig. 2c). Surprisingly, the iCP α-ring pocket occupancy status in our mammalian PA28αβ-iCP differs considerably from that described for PfPA28-cCP (Supplementary Table 4), which showed only one weak PfPA28 C-terminal insertion into the α7/1 pocket of cCP although it showed a slight tilt of the activator toward α3–α4[42]. In contrast, the iCP α-ring occupancy status of PA28αβ-iCP differs only a little from that of TbPA26-cCP, with the only major difference being a lack of a PA26 C terminus insertion into the α1/2 pocket of the cCP (Supplementary Table 4)[43]. Taken together, our data suggested the binding mode and association mechanism between the heteroheptameric PA28αβ and iCP (and most likely also for cCP) to be quite different from that described for the homoheptameric PfPA28 with cCP in terms of strength and location, but to be more comparable to that of TbPA26 with cCP.

Furthermore, another critical contact between PA26 and cCP has been indicated to involve the formation of interactions between the activation loop of PA26 and the reverse turn of the CP α-subunit (located between the N-terminal tail and H0 of α-subunit), and in part lead to the gate opening of the CP[41,43,59]. Indeed, the TbPA26-cCP structure shows interactions between all of the activation loops of the homoheptameric PA26 and the related reverse turns of CP[41]. In our PA28αβ-iCP structure, all seven activation loops of PA28αβ were resolved, with the ones interacting with α1, α3, α4, and α6 of iCP showing stronger pieces of density (Fig. 3c and Supplementary Table 5). Moreover, our binding assay and the proteolytic activity assay together showed that the activation loop double mutated PA28α$^{N146Y}$β$^{N136Y}$ can still bind iCP (Supplementary Fig. 5c), but failed to activate iCP; while the only PA28β mutated PA28αβ$^{N136Y}$ decreased the activation of iCP by about half (Supplementary Fig. 5d), in agreement with previous biochemical studies on the PA28 activator[40,60]. Collectively, our data indicate that the activation loops especially N146 in PA28α and N136 in PA28β are the key structural elements for the activation and gate opening of iCP. In contrast, the activation loop-reverse turn interaction in the PfPA28-cCP structure was described to be very different, and showed interactions only on the α6–α7 side of CP (including α5, α6, α7, and α1), with the interactions with α6 and α7 appearing stronger, but did not reveal the activation loop densities interacting with the gatekeeper α2/3/4 subunits (Supplementary Table 5)[42].

In addition, we also observed conformational changes in iCP induced by PA28αβ binding, including a slight shift (up to 2.3 Å) in the reverse turn regions of the gate keepers α2/3/4 and the neighboring α5 in iCP (Fig. 3d). Similar phenomenon had been well documented in the PA26/PAN/Blm10 systems, suggesting that minor conformational changes in reverse turns induced by activator binding could disturb the allosteric network in the gate region, and lead to the gate opening of CP[41,43,61,62]. In addition, we also found slight rotations of the peripheral portions of α3 and α5 surrounding the α4-subunit; these rotations could play a role in

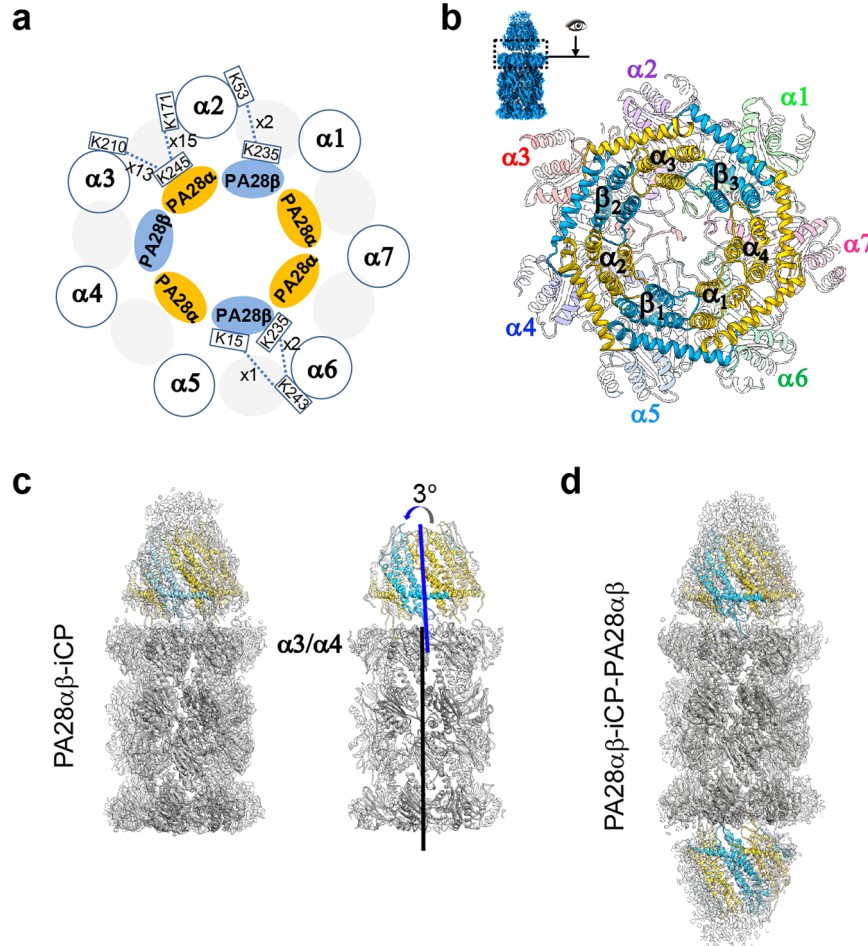

**Fig. 2 The spatial arrangement between PA28αβ and iCP in the PA28αβ-iCP proteasome. a** A carton diagram illustrating the spatial arrangement between PA28αβ and iCP (blue circles) derived from XL-MS analysis. The PA28α and PA28β are shown as sky blue and gold ellipses, respectively, and the color scheme was followed throughout. Identified cross-linked contacts (only in the interface) between a pair of subunits are shown as dotted lines, with the involved residues and the spectrum count number indicated. We used an *E*-value of 1.00E−02 as the threshold to remove extra lower-confidence XL-MS data. **b** Visualization of the PA28αβ-iCP interface, with the iCP subunits shown in distinct colors and appearing transparent. The visualization angle and region are illustrated in the inset. **c** Cryo-EM map of PA28αβ-iCP (transparent gray) with fitted model (iCP in gray, PA28αβ in color, left panel). On the right showing the model of PA28αβ-iCP, with the position of the α3/α4-subunit and the axes of PA28αβ (blue) and iCP (black) indicated. The axes also indicate the direction of the tilt of PA28αβ relative to iCP. **d** Cryo-EM map and model of PA28αβ-iCP-PA28αβ.

disturbing the allosteric network of the gate keepers α2/3/4 (and to a lesser extent of α5) (Fig. 3e). These motions may have been induced by the noticeable leaning of PA28αβ toward the α3-α4 side of the iCP, and could be propagated to the β-ring with visible movements of the underneath β4 and β6 subunits (Fig. 3e). In contrast, *Pf*PA28 binding was indicated to not induce any large changes in the conformations of the α-ring subunits of the cCP[42]. While for the *Tb*PA26-cCP complex, *Tb*PA26 binding was indicated to result in a rearrangement of the N-terminal extensions of α2, α3, α4, and α5 to a conformation similar to that of α6, α7, and α1 and in this way to an opening of the axial pore[41].

Taken together, our data indicated that, relative to the homoheptameric *Pf*PA28 or *Tb*PA26, the heteroheptameric mammalian PA28αβ interacts with and activates the enzymatic CP using related but different mechanisms, especially distinct from that of *Pf*PA28; consequently, the motions they induced to CP are also divergent. These observations, along with the relative low sequence identities between these proteins from different species (Supplementary Fig. 6), suggested that during evolution the heteroheptameric PA28αβ has undergone profound remodeling to achieve its current physiological functions including immune response in eukaryotes.

**The mechanism of PA28αβ-induced partial gate opening of the iCP.** Activation of proteasomes largely relies on the gate opening of CP, which could be triggered by regulators or together with substrates[36,41,43,62]. Our free iCP map showed a closed-gate configuration with ordered and well-resolved N termini of the α2, α3, α4, and to a lesser extent α5 subunits showing extended conformations covering the gate region (Fig. 4a); while the N termini of the remaining α6, α7, and α1 subunits were observed to point away from the pore and to approximately align with the central axis of the iCP, thus not participating in blocking the iCP pore. In contrast, in our PA28αβ-iCP map, the N termini of the α2, α3, and α4 subunits in the PA28αβ-contacting iCP α-ring appeared disordered, suggesting that the binding of the PA28αβ activator to iCP caused these regions to become more dynamic (Fig. 4b). This plasticity in these N-terminal regions may have disrupted the allosteric networks in the gate region, leading to a partially open gate in the contacting α-ring, while leaving the gate in the opposite α-ring closed (Fig. 4b). Consistent with this proposal, both gates appeared to have adopted a partially open conformation in our double-capped PA28αβ-iCP-PA28αβ map (Fig. 4c). Thus, the binding of the PA28αβ activator to the iCP was concluded to have induced a partial gate opening of the iCP.

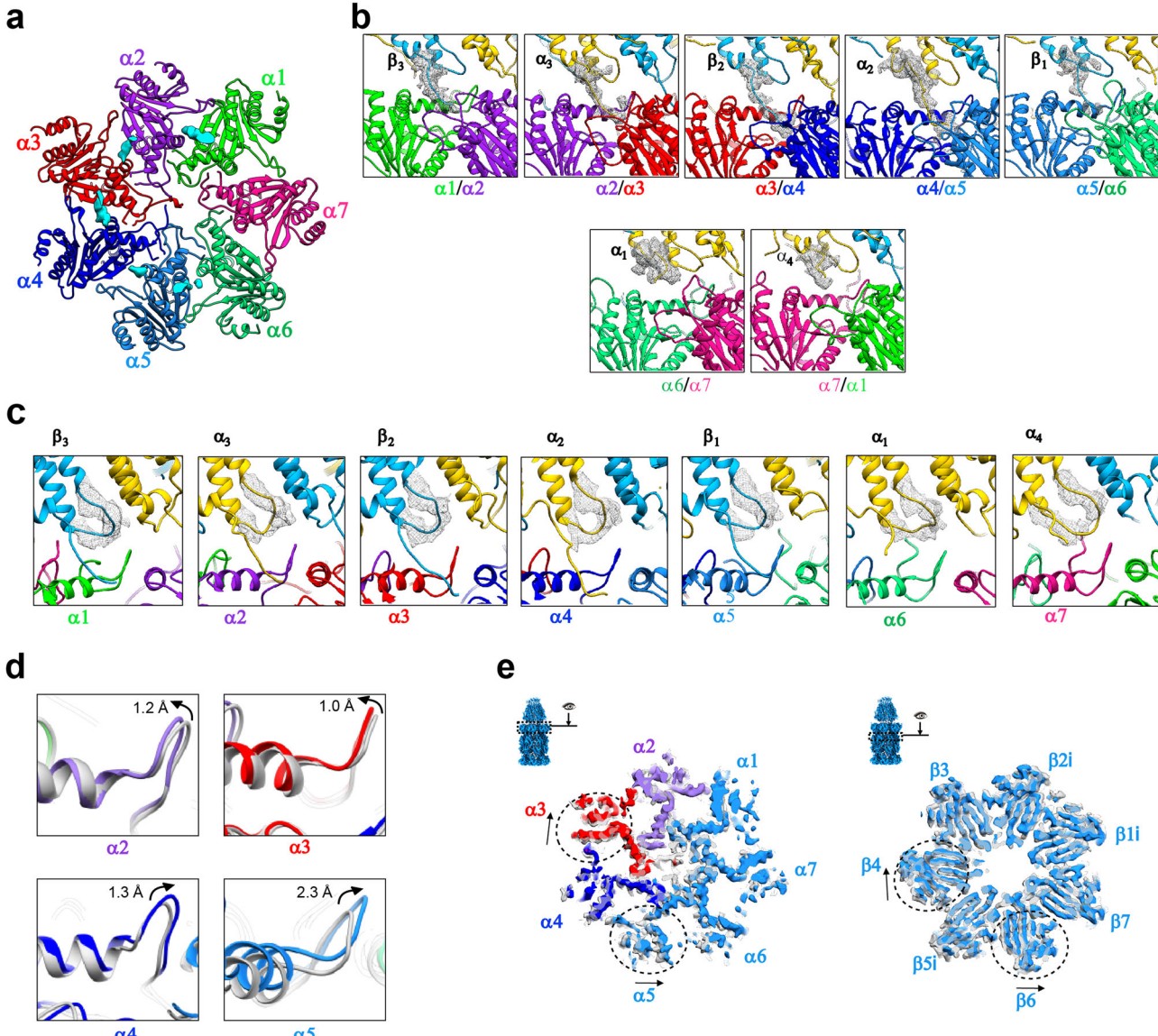

**Fig. 3 Major interactions between PA28αβ and iCP and the induced conformational changes of iCP. a** Top view of the iCP α-ring with the inserted C-terminal tails of PA28αβ (cyan pieces of density) for the PA28αβ-iCP complex. **b** Magnified views of the insertions of PA28αβ C termini into the iCP α-ring pockets for the PA28αβ-iCP complex, with the C-tail density shown as gray wire. The color schemes are as in Fig. 2b and are followed throughout. **c** The activation loop status of PA28αβ (the loop density shown as gray wire) in the PA28αβ-iCP complex. **d** Conformational changes of the reverse turns induced by PA28αβ association in the PA28αβ-iCP complex, determined by aligning the models of PA28αβ-iCP (in color) with that of the free iCP (gray). Black arrow indicates the shift direction. **e** Superpositions of the *cis* α-rings (left) and β-rings (right) of PA28αβ-iCP (with color) and the free iCP (gray), showing their conformational differences. Dashed circles indicate the regions with considerable conformational changes between the two structures, and arrows show the direction of conformational switches. The visualization angle and region are illustrated in the inset.

In general, there are two important elements of proteasome activation by 11S activators: the C termini of the activator, which provide binding energy; and the activation loop-reverse turn interaction that can destabilize the closed-gate conformation[41]. Here, by comparing the structure of the activator-associated PA28αβ-iCP complex with that of free iCP, we derived a detailed proposed mechanism for the iCP partial gate opening induced by the mammalian PA28αβ activator (Fig. 4d). According to this mechanism, the binding of PA28αβ with iCP would associate with the insertion of the C-terminal tails of four consecutive PA28αβ subunits (β₃, α₃, β₂, and α₂) into the corresponding α-pockets (α1/2, α2/3, α3/4, and α4/5), causing PA28αβ to lean towards the α3–α4 side of iCP. The close proximity in this position could facilitate the formation of interactions between the activation loops of PA28αβ and the reverse turns of the iCP

subunits, especially the α2/3/4 gate keepers, resulting in a shift in their reverse turn regions. As the N termini of α1, α6, and α7 have been observed to adopt conformations pointing away from the proteasome, they would contribute less to the gate formation; while α2/3/4, and to a lesser extent α5, have been observed to pack closely to cover the gate. Thus, the interactions between the reverse turns of α2/3/4 and the corresponding PA28αβ activation loops (especially N146 in PA28α and N136 in PA28β) would disrupt the allosteric networks of the gate keepers, resulting in a partially open gate of iCP in the PA28αβ-iCP complex.

**Unique properties of the beta catalytic subunits in bovine immunoproteasomes.** During infection of antigens, PA28αβ as well as the β1i, β2i, and β5i subunits of the CP would be induced

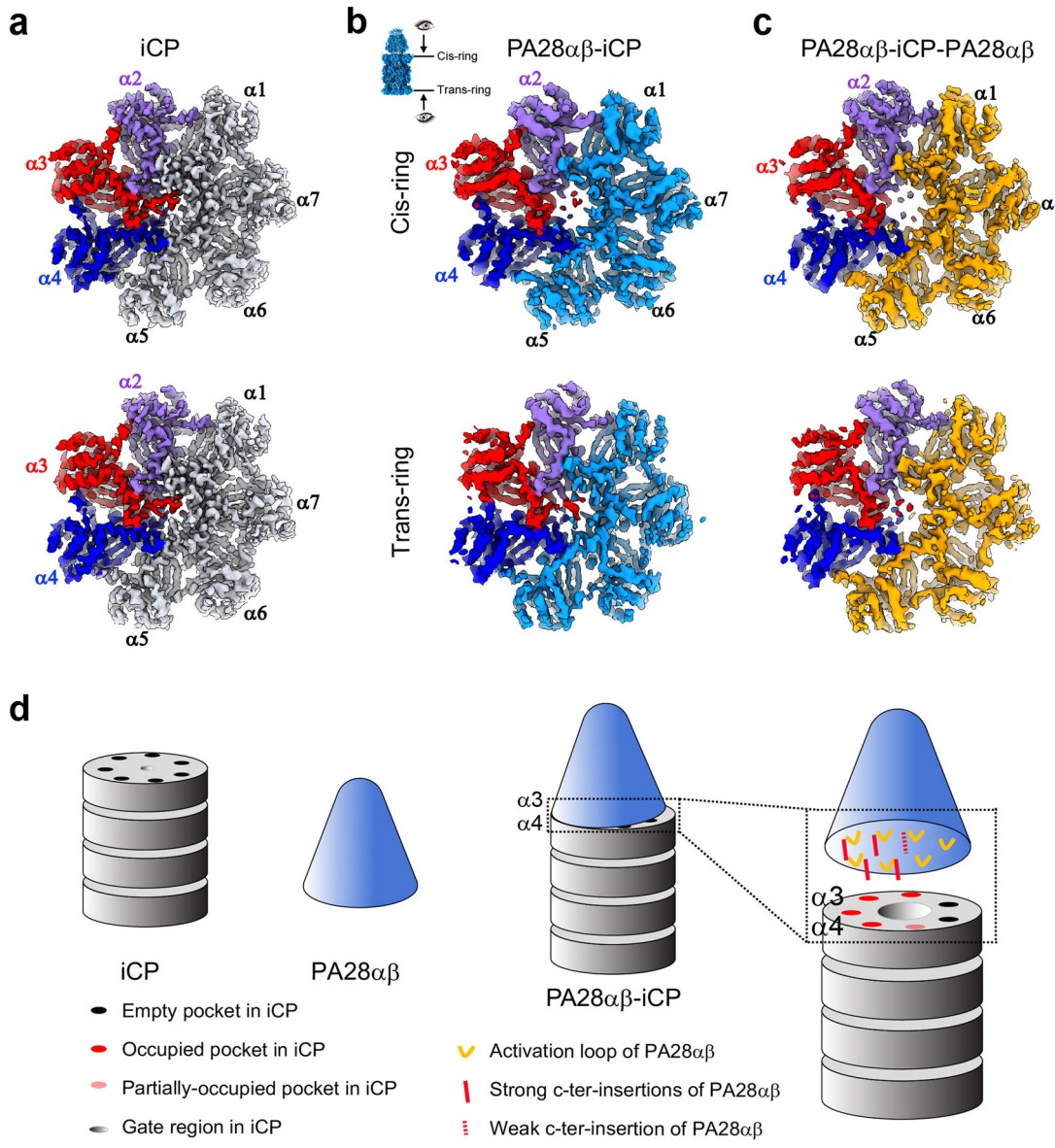

**Fig. 4 The iCP gate status in bovine immunoproteasomes and a proposed gate opening mechanism induced by PA28αβ binding. a** The gate status of the free bovine iCP, with both rings showing a closed-gate conformation. The gatekeeper subunits α2, α3, and α4 are shown in purple, red, and blue, respectively, which scheme is followed throughout this figure. **b** The gate status of bovine PA28αβ-iCP, with the PA28αβ-capped *cis*-ring showing a partially open gate, while the *trans*-ring still showing a closed gate. The visualization angle and region are illustrated in the inset. **c** The gate status of PA28αβ-iCP-PA28αβ, with both rings showing a partially open gate. **d** A schematic diagram depicting a proposed mechanism of gate opening induced by PA28αβ binding.

by IFN-γ to form immunoproteasomes, facilitating the generation of MHC class I ligands for subsequent antigen presentation[26,27,63]. However, the underlying molecular mechanism responsible for the stimulation of immunoproteasome activity remains unclear. Here, our subunit alignment analysis suggested that the association of PA28αβ with iCP would tend to have a subtle effect on the conformations of β1i and β2i, although it could slightly reshape a turn located outside the chamber and helix 3 (H3) of β5i (located in the outermost region of the chamber) (Supplementary Fig. 7a). Besides, after association with PA28αβ the electrostatic surface of β1i and β2i proteolytic sites exhibited slightly weaker positive electrostatic potential compared with that of free iCP; also, in β2i a surrounding loop with negative charge properties appeared moving inward toward the proteolytic site relative to the free iCP; while for β5i, there was not much obvious changes in surface property (Supplementary Fig. 7b). The subtle conformational changes and

surface property variations induced by PA28αβ association may contribute to the regulation of iCP immunoproteasome activity, especially for β1i and β2i. Still, the production of antigen ligands in the iCP may mostly arise from the replacement of standard β subunits with immune β subunits.

We then compared the conformations of the three catalytic subunits of our free bovine iCP structure with those of the available bovine cCP structure (PDB ID: 1IRU)[64]. This comparison revealed noticeable conformational differences between them in several regions (Fig. 5a). For instance, relative to cCP, in iCP the C-terminal tail of β1i and a linker region (Gly133-Leu139) both showed conformational rearrangement, the C-terminal loop of β2i exhibited a slight outward displacement, and the β5i H3 displayed an observable outward shift (Fig. 5a). Interestingly, we observed similar conformational differences between the human iCP and cCP and between the mouse iCP and cCP (Supplementary Fig. 8a, b).

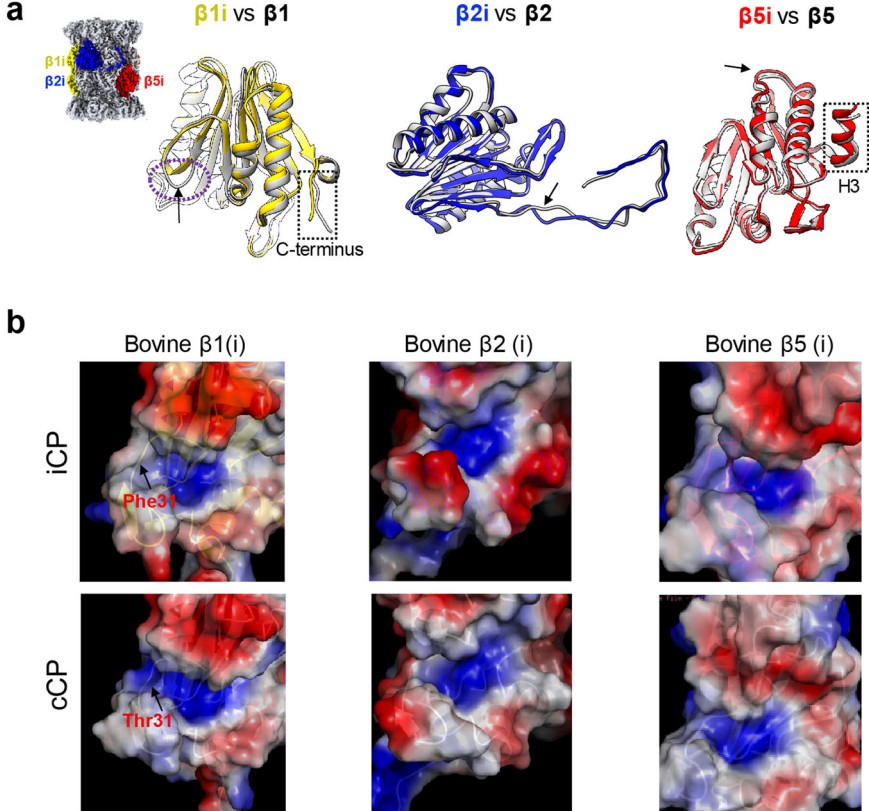

**Fig. 5 Structural comparison of the bovine iCP with bovine cCP. a** Structural superpositions of the three catalytic subunits of the free bovine iCP (with β1i, β2i, and β5i colored yellow, blue, and red, respectively) on the corresponding subunits of the bovine cCP (in gray, PDB ID: 1IRU). The observable conformational changes are indicated by black arrow and dotted ellipsoid or rectangle, which are followed throughout. **b** Surface property representations of the catalytic pockets of the three enzymatic subunits for free bovine iCP and bovine cCP, with the most distinct residues between β1i and β1 in this region indicated.

These data suggested that the conserved conformational changes and related elements may play a role in immunoproteasome activity stimulation. Furthermore, for bovine proteasome, the patterns of surface property in the catalytic pockets of β1i were observed to be different from those of β1 (Fig. 5b), with this difference mainly resulting from the difference between β1-Thr31 and β1i-Phe31. Interestingly, a similar amino-acid residue and corresponding surface property difference were also observed between human β1i and β1, as well as between mouse β1i and β1 (Supplementary Fig. 8c), indicating these conserved differences may contribute to the suppression of caspase-like activity in β1i in mammalian iCPs[65]. Collectively, these findings may to some extent facilitate our understanding of the mechanism of immunoproteasome activity stimulation.

## Discussion

Unlike 19S activator, the 11S activators (PA28/REG and PA26) stimulate the degradation of substrates in an ATP/ubiquitin-independent manner. Within the 11S family, PA28αβ forms a heteroheptamer while the remaining members (PA28γ and PA26) form respective homoheptamers. PA28αβ plays an important role in MHC class I antigen presentation[29,66]. To date, there remains a lack of structural information on the intact mammalian PA28αβ-cCP or PA28αβ-iCP complex, and thus the mechanism by which PA28αβ binds and activates iCP is elusive. Here we provide insight into these mechanisms by acquiring and analyzing the cryo-EM structure of mammalian PA28αβ-iCP immunoproteasome complex with degradation ability as well as the structure of free bovine spleen iCP (Fig. 1). We determined

the spatial arrangement between PA28αβ and iCP units relative to each other (Fig. 2a), which allowed us to derive a mechanism for the interaction and activation of iCP by the heteroheptameric PA28αβ, largely distinct from the mechanisms previously proposed for cCP by the homoheptameric TbPA26 or PfPA28. Our findings suggest that PA28αβ-iCP has experienced a profound remodeling during evolution to achieve its current level of function in immune response.

**A distinct proteasome activation mechanism adopted by mammalian PA28αβ.** Although human PA28αβ, PfPA28, and TbPA26 are all 11S activators, they appear to cause CP gates to be opened to different extents. In contrast to the fully open gate of the CP in TbPA26-CP, here our structural study demonstrated that human PA28αβ binding could cause the formation of a partially open gate of the iCP (Fig. 4). This difference in the CP gate opening status induced by the two proteasome activators could be attributed to several factors: (1) their different oligomerization states, with human PA28αβ being heteroheptameric but TbPA26 homoheptameric; (2) their different orientations relative to the CP, with PA28αβ being slightly tilted, but TbPA26 untilted; and (3) their different interactions with the CP, with the binding of human PA28αβ to iCP perhaps being overall stronger (with one more C-terminal insertion), but the activation loop-reverse turn interactions of TbPA26 appearing more evenly distributed and stronger (Supplementary Tables 4 and 5), which could facilitate the full opening of the CP gate.

Moreover, although the PfPA28-CP structure in a recent study also showed a slight leaning of the activator to the α3–α4 side and

a partially open CP gate[42], similar to that observed in our PA28αβ-iCP structure, the mode and strength of interaction between *Pf*PA28 and CP appeared very different from that of PA28αβ with iCP. For instance, human PA28αβ showed five C-terminal tail insertions into the α-ring pockets around the α3–α4 side (Fig. 3a, b and Supplementary Table 4), while *Pf*PA28 displayed much weaker interaction with only one insertion into the opposite side (the α7/1 pocket) (Supplementary Table 4). Moreover, our PA28αβ-iCP structure allowed us to capture all seven activation loops stabilized by the interaction with the reverse turn of the α-subunits (albeit with three of them being relatively dynamic) (Fig. 3c and Supplementary Table 5). The *Pf*PA28-cCP structure showed four such interactions only on the α6–α7 side of the CP (with two of them appearing stronger), and no density were observed corresponding to activation loops interacting with the α2/3/4 gate keepers (Supplementary Table 5)[42]. Taken together, the mammalian PA28αβ may use a mechanism distinct from those of *Pf*PA28 and *Tb*PA26 to activate the enzymatic CP. This is in line with a recent Hydrogen–Deuterium eXchange coupled to Mass Spectrometry study indicating that different sets of 20S α-subunit N terminus were destabilized depending on the PA28/20S pair[67].

**A potential on-and-off mode to regulate the CP gate opening by PA28αβ.** Compared with the non-ATPase PA28αβ activator, the interaction and activation mechanism exercised by the AAA + ATPase 19S activator has been indicated to be more precisely regulated and to involve different structural elements, with the C-terminal tails of several Rpt subunits from the hexameric ATPase ring inserted into the related α-ring pockets of CP, without direct interactions formed by activation loops and reverse turns as in the case of PA28αβ. Also, the number of inserted Rpt C-terminal tails has been indicated to vary during its ATP cycle, with an extra Rpt6 or Rpt6/1 C-tail insertion (into the pockets in the α2/3/4 side) used to trigger the opening of the 20S gate[37,39,68]. In contrast, for PA28αβ-iCP, we showed that the slight leaning of the PA28αβ relative to the iCP could lead to a more intimate interaction with the gatekeeper α2/3/4 subunits (Fig. 3), facilitating the gate opening and activation of the iCP. Interestingly, although the activation seems to occur through different structural elements, they all regulate the allosteric network of the same enzymatic machine, i.e., the 20S CP, and eventually disturb the allosteric network in the key α2/3/4 gate keepers to trigger the gate opening, readying the proteasome for substrate translocation and processing within the enzymatic core. This suggests that the 20S CP has evolved to be complex enough that uses one set of allosteric network to take signals from diversified activators; still, the sophisticated machine can eventually collect all the signals and converge them to touch the ultimate key trigger points to activate the entire assembly.

Furthermore, for the ATPase-containing 19S activator, binding is not sufficient to trigger the CP gate opening; instead, it requires ATP to eventually drive the CP gate status varying between closed and open facilitating substrate processing[37,38]. In contrast, for the non-ATPase PA28αβ activator, it appears that the binding and activation could happen simultaneously, i.e., once the PA28αβ activator associates with the CP, the gate of CP would open (Fig. 4b, c). Also given the difficulty of a steady association between PA28αβ and CP observed here as well as in a previous study[45], we postulated that the non-ATPase PA28αβ activator likely uses an on-and-off mode (association and disassociation with the CP) to regulate CP gate opening and closing and substrate processing, as may also be the case for other non-ATPase proteasome activators. This may reflect that, during the course of evolution, the physiological role of the activator and the divergent level of the substrates determine the structural complexity of the activator and the mechanism of CP gate regulation.

**Mechanism of enzymatic activity stimulation of bovine immunoproteasome.** The formation of immunoproteasomes induced by IFN-γ could enhance proteasomal trypsin-like (β2) and chymotrypsin-like (β5) activities and suppress caspase-like activity (β1)[69–71], facilitating the generation of MHC class I ligands for subsequent antigen presentation. In addition, IFN-γ-induced expression of PA28αβ could markedly stimulate proteasomal degradation of short peptides in vitro[45,72], leading to substantial changes in the pattern of peptides generated[66]. However, little is known about the molecular mechanism of the immunoproteasome activity stimulations.

Here we found some noticeable conserved conformational differences in several loops of β1i and β2i, and the H3 helix of β5i between the iCP and cCP in bovine, human, and mouse (Fig. 5 and Supplementary Fig. 8a, b), and these differences might play a role in immunoproteasome functions. Also, we observed a surface property difference in the catalytic center of β1i distinct from that of β1 (Fig. 5b), mainly resulting from the primary sequence variation between β1i and β1 in the enzymatic pocket. Interestingly, this phenomenon was conserved among human, mouse, and bovine (Supplementary Fig. 8c), which could contribute to the lower caspase-like (β1) activity of the immunoproteasome[69–71]. Collectively, the variant amino-acid residues and resulting distinct surface properties may contribute to the regulation of iCP immunoproteasome activity, especially for β1i.

**Potential effect of PA28αβ on substrate processing.** Substrates have been proposed to go through the central channel running along PA28αβ to enter the chamber of the CP[73]. Also, PA28αβ may control the efflux of longer peptides out of the proteolytic chamber and contribute to their ongoing hydrolysis. Hence PA28αβ could serve as a selective sieve that controls the entry of substrate and/or the exit of degradation products[42]. The quite narrow channel entrance (20 Å) of PA28αβ may to some extent impose a stretching or unfolding force on the engaged substrates/products to regulate their entry/exit (Fig. 1a). Besides, our structure also allowed us to visualize the substrate-recruitment loops of PA28αβ, which tend to form a dome loosely covering the entrance of the PA28αβ channel (Fig. 1a, b). The intrinsic plasticity of these loops may be beneficial for their involvement in the substrate recruitment and selection or exit of the intermediate/final product.

In summary, our study has revealed the complete architecture of mammalian immunoproteasome PA28αβ-iCP and that of the bovine iCP, and has provided insights into a distinct mechanism by which PA28αβ might activate the proteasome. Our data have also partially clarified the regulation mechanism of iCP immunoproteasome activity, especially for β1i, beneficial for inhibitor development. Interestingly, we delineated a mechanism on how the sophisticated 20S CP has evolved to use one set of allosteric networks to take signals from diversified activators, and eventually converge these signals to touch the ultimate key trigger points to open the gate. We also proposed an on-and-off mode by which the non-ATPase activator such as PA28αβ likely uses to regulate CP gate opening and closing, and provided insights in the potential effect of PA28αβ on substrate processing.

## Methods
**Molecular biology and purification of PA28αβ and mutated PA28αβ.** For co-expression of human PA28α and PA28β, their DNA fragments were inserted into the pETDuet-1 vector system (MSC1: N-terminal-6xHis-PA28α; MSC2: N-

terminal-Flag-PA28β). Reconstructed plasmids were transformed into *E. coli* BL21 (DE3) cells and selected for resistance to ampicillin on agar plates. Transformants were grown at 37 °C in liquid LB medium supplemented with ampicillin. Gene expression was induced using 1 mM IPTG. After induction for 16 hours (h) at 18 °C, cells were harvested and frozen at −80 °C. The frozen *E. coli* cells were lysed with an ultra-high-pressure cell disrupter in lysis buffer (50 mM Tris-HCl pH 7.5, 100 mM NaCl, 1 mM DTT, 10% glycerol). The lysate was centrifuged at 20,000 *g* for 30 min at 4 °C. The clarified lysate was incubated with Ni-NTA agarose beads (Sigma) for 30 min at 4 °C. The beads were recovered and washed twice with wash buffer (50 mM Tris-HCl pH 7.5, 100 mM NaCl, 1 mM DTT, 10% glycerol, 20 mM imidazole) before eluting with 50 mM Tris-HCl, pH 7.5, 100 mM NaCl, 10% glycerol, 200 mM imidazole. The collected elution was subsequently incubated with anti-FLAG M2 agarose beads (Sigma) for 2 h at 4 °C. Then, the beads were recovered and washed twice with wash buffer (50 mM Tris-HCl pH 7.5, 100 mM NaCl, 10% glycerol) before eluting with 500 μg/mL 3× FLAG peptide (Shanghai Biotech BioScience & Technology). The resulting samples were concentrated by using 100 kDa centrifugal filter devices and subjected to size-exclusion chromatography (Superdex 200 16/60; GE Healthcare) with 20 mM Tris-HCl pH 7.5, 100 mM NaCl, and 1 mM DTT. The fractions were further analyzed using SDS-PAGE, western blot, and MS (Supplementary Fig. 1a). For long-term storage, the PA28αβ samples were supplemented with 15% glycerol and stored at −80 °C. The plasmids of the mutated PA28αβ^ΔC-tails^ (truncated K245-Y249 for PA28α and E234-Y239 for PA28β), PA28αβ^N136Y^ (only N136Y mutation in PA28β), and PA28α^N146Y^β^N136Y^ (N146Y mutation in PA28α and N136Y mutation in PA28β) were reconstructed based on our above-described pETDuet-PA28αβ plasmid. These mutated PA28αβ proteins were purified respectively following the same purification process of normal PA28αβ.

**Purification of bovine iCPs.** Bovine iCPs were purified from bovine spleen following a procedure similar to that described previously[74]. Bovine spleen (50 g) was cut into pieces and homogenized in lysis buffer (25 mM Tris-HCl pH 7.4, 10 mM MgCl₂, 4 mM ATP, 1 mM DTT, 10% glycerol). The homogenate was centrifuged at 20,000 *g* for 30 min and then centrifuged at 100,000 *g* for 1 h to remove cell debris and membranes. The supernatant was applied to a 100 mL DEAE-Affigel Blue (Bio-Rad) column equilibrated with buffer A (25 mM Tris-HCl pH 7.4, 10 mM MgCl₂, 1 mM ATP, 1 mM DTT, 10% glycerol). Then the resin was washed with buffer A and 50 mM NaCl in buffer A. Proteasomes were eluted with buffer AN (25 mM Tris-HCl pH 7.4, 10 mM MgCl₂, 1 mM ATP, 1 mM DTT, 10% glycerol) and directly applied to a 50 mL Source 15Q column equilibrated with buffer AN. The source 15Q column was washed with buffer AN and eluted with a 500 mL gradient of 150–500 mM NaCl in buffer A. Proteasomes were eluted at a salt concentration of 300–330 mM NaCl and the activities of the collected fractions were monitored by performing a peptidase activity assay (described below). Fractions containing proteasomes were desalted and concentrated, and finally purified by using a glycerol gradient (15–45% glycerol (wt/vol), 25 mM Tris-HCl pH 7.4, 10 mM MgCl₂, 1 mM DTT) and subjecting them to centrifugation for 16 h at 234,700 *g*. Note that the fractions containing iCP were identified by performing SDS-PAGE and negative-staining EM, as well as the peptidase activity assay (Supplementary Fig. 1b, e). The samples were frozen using liquid nitrogen and stored at −80 °C.

**In vitro reconstitution.** Before mixing together the above-purified PA28αβ and iCP, PA28αβ was dialyzed against buffer1 (20 mM Hepes pH 7.5, 100 mM NaCl, 10% glycerol, and 1 mM DTT), and iCP was dialyzed against buffer2 (25 mM Hepes pH 7.4, 10 mM MgCl₂, 1 mM DTT, 10% glycerol). The dialyzed PA28αβ was mixed with the dialyzed iCP at a molar ratio of 10:1 and incubated at 37 °C for 30 min. Then glutaraldehyde, at a final concentration of 0.1% (vol/vol), was added into this system, which was further incubated at 4 °C for 2 h. Tris-HCl pH 7.4 at a final concentration of 50 mM was then added to this system to terminate the glutaraldehyde-induced cross-linking reaction. Finally, the reconstituted sample was purified using a glycerol gradient (15–45% glycerol (wt/vol), 25 mM Tris-HCl pH 7.4, 10 mM MgCl₂, 1 mM DTT) and subjected to centrifugation for 16 h at 234,700 *g*. Fractions containing PA28αβ-iCP proteasomes were identified by peptidase activity assay and EM analysis (Supplementary Fig. 1c–e).

**Peptidase activity assay.** The activity of the proteasome was monitored by performing a peptidase activity assay as previously described[74]. To quickly determine which fraction or fractions contain proteasomes, we added Suc-LLVY-AMC to a 96-well plate for each sample. The 96-well plate was incubated at 37 °C and visualized by using a gel image system (Tanon-1600, Shanghai, China). We continuously measured the proteasome proteolytic activity by continuously monitoring the fluorescence of free AMC for 10 min using a multimode microplate reader (BioTek).

**Cryo-EM sample preparation and data collection.** Holey carbon grids (Quantifoil R2/1, 200 mesh) were plasma treated using a Solarus plasma cleaner (Gatan). A volume of 2 μL of the sample was placed onto a grid, then flash frozen in liquid ethane using a Vitrobot Mark IV (Thermo Fisher). Data collection was performed using a Titan Krios transmission electron microscope (Thermo Fisher) operated at

300 kV and equipped with a Cs corrector. Images were collected by using a K2 Summit direct electron detector (Gatan) in super-resolution mode (yielding a pixel size of 1.32 Å after 2 times binning). Each movie was dose-fractioned into 38 frames. The exposure time was 7.6 s with 0.2 s for each frame, generating a total dose of ~38 e⁻/Å². All of the data were collected using the SerialEM software package[75] with defocus values ranging from −1.5 to −2.8 μm.

**Cryo-EM 3D reconstruction.** A total of 3170 micrographs were used for the structure determinations. All images were aligned and summed using Motion-Cor2[76]. Unless otherwise specified, single-particle analysis was mainly executed in RELION 3.0[77]. After CTF parameter determination using CTFFIND4[78], particle auto-picking, manual particle checking, and reference-free 2D classification, 274,210 particles remained in the dataset (Supplementary Fig. 2). The initial model was a single-capped PA26-20S proteasome, derived from the previous crystal structure (PDB: 1Z7Q)[43], and low-pass filtered to 60 Å using EMAN 1.9[79].

For the reconstruction, one round of 3D classification was carried out and resulted in extraction of 28% good PA28αβ-iCP particles and 60% good free iCP particles. Then another round of 3D classification was performed to further separate double-capped PA28αβ-iCP-PA28αβ and single-capped PA28αβ-iCP particles. An auto-refine procedure was performed in RELION for each of the three kinds of particles to generate their corresponding maps. Afterwards, particles were sorted by carrying out multiple rounds of 3D classifications, yielding a PA28αβ-iCP dataset of 45,030 particles, a PA28αβ-iCP-PA28αβ dataset of 24,627 particles, and a free iCP dataset of 56,663 particles. These particles were re-centered and polished, and one more round of auto-refine procedure was performed, resulting in a 4.1-Å-resolution map of PA28αβ-iCP, a 4.2-Å-resolution map of PA28αβ-iCP-PA28αβ, and a 3.3-Å-resolution map of free iCP. These maps were sharpened by applying corresponding negative B-factors, estimated by using an automated procedure in RELION 3.0. For each of the reconstructions, the resolution was accessed based on the gold-standard criterion of FSC = 0.143, and the local resolution was estimated by using ResMap[80].

**Pseudo-atomic-model building.** We used the X-ray crystal structures of the bovine cCP (PDB: 1IRU)[64] and mouse PA28αβ (PDB: 5MX5)[40] as template to build the homology models of the bovine iCP and human PA28αβ, respectively, through the SWISS-MODEL webserver[81]. The well-resolved sidechain densities throughout our bovine free iCP map and the iCP portion within the map of PA28αβ-iCP enabled us to amend and refine the entire atomic model. This refinement was carried out first using COOT[82], then using the *phenix.real_space_refine* program in Phenix[83]. The final atomic model was validated using *phenix.molprobity*. After determining the relative spatial arrangements between PA28αβ and iCP, we also built a pseudo-atomic model for the PA28αβ-iCP complex following the same procedure. In addition, the C-terminal tails of PA28αβ, which were not included in the initial homolog model, were resolved in our PA28αβ-iCP map. We then added and refined these tails using COOT[84] and Phenix against the corresponding map density. The validation statistics for the atomic models are summarized in Supplementary Table 2. Figures were generated with either UCSF Chimera or ChimeraX[85,86], as well as PyMOL (http://www.pymol.org).

**Cross-linking/mass spectrometry analysis.** We mixed the individually purified and dialyzed PA28αβ and iCP at a molar ratio of 10:1 and incubated them at 37 °C for 30 min. We then cross-linked the associated PA28αβ-iCP complex by bis(sulfosuccinimidyl) suberate (BS3), with a final cross-linker concentration of 0.25 mM. After incubation on ice for 2 h, 20 mM Tris-HCl was used to terminate the reaction. To remove un-cross-linked components in the reconstituted sample, we performed further glycerol gradient centrifugation (15–45% glycerol (wt/vol), 25 mM Tris-HCl pH 7.4, 10 mM MgCl₂, 1 mM DTT) for 16 h at 234,700 *g*. We identified the fractions containing PA28αβ-iCP proteasomes by performing peptidase activity assay and NS-EM. Cross-linked complexes were buffer exchanged again to Hepes buffer with 1% glycerol and then subjected to MS experiment. The proteins were precipitated and digested for 16 h at 37 °C by trypsin at an enzyme-to-substrate ratio of 1:50 (w/w). The tryptic digested peptides were desalted and loaded on an in-house packed capillary reverse-phase C18 column (40 cm length, 100 μM ID × 360 μM OD, 1.9 μM particle size, 120 Å pore diameter) connected to an Easy LC 1200 system. The samples were analyzed with a 120 min-HPLC gradient from 6 to 35% of buffer B (buffer A: 0.1% formic acid in Water; buffer B: 0.1% formic acid in 80% acetonitrile) at 300 nL/min. The eluted peptides were ionized and directly introduced into a Q-Exactive mass spectrometer using a nanospray source. Survey full-scan MS spectra (from m/z 300–1800) were acquired in the Orbitrap analyzer with resolution *r* = 70,000 at m/z 400. Cross-linked peptides were identified and evaluated using pLink2 software[87]. Free iCP sample was cross-linked under 2 mM BS3 and the XL-MS experiment was conducted following similar procedure.

## Data availability
Data supporting the findings of this manuscript are available from the corresponding author upon reasonable request. EM maps of iCP, PA28αβ-iCP, and PA28αβ-iCP-PA28αβ have been deposited in the Electron Microscopy Data Bank under accession codes EMDB-30825, EMDB-30824, and EMDB-30828, respectively. Pseudo-atomic

models for iCP, PA28αβ-iCP, and PA28αβ-iCP-PA28αβ have been deposited in the Protein Data Bank under accession numbers of 7DR7, 7DR6, and 7DRW, respectively. Source data are provided with this paper.

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

## Acknowledgements

We are grateful to the staffs of the NCPSS Electron Microscopy facility, Database and Computing facility, Protein Expression and Purification facility, and Mass Spectrometry facility for instrument support and technical assistance. This work was supported by grants from the NSFC-ISF 31861143028, the Strategic Priority Research Program of CAS (XDB37040103), National Basic Research Program of China (2017YFA0503503), the NSFC (31670754 and 31872714 to Y.C, 31800623 to Z.D.), Shanghai Academic Research Leader (20XD1404200), and the CAS Facility-based Open Research Program and the CAS-Shanghai Science Research Center (CAS-SSRC-YH-2015-01, DSS-WXJZ-2018-0002).

## Author contributions

J.C., Y.W., and Y.C. designed the experiments. J.C. and Y.W. purified the proteins, performed functional analysis, and collected the cryo-EM data. J.C., Z.D., and Y.W. performed data reconstruction. C.X. and J.C. performed model building. Y.W., K.C., Q.Z., and S.W. performed the mutation and truncation experiments. C.P., Y.Y., C.X., and J.C. performed the XL-MS experiments and data analysis. J.C., Z.D., and Y.C. analyzed the structure and wrote the manuscript.

## Competing interests

The authors declare no competing interests.
