## [Peer Review File · Nature Communications]

REVIEWER COMMENTS

Reviewer #1 (Remarks to the Author):

In this work entitled Cryo-EM of mammalian PA28 $\alpha\beta$ -iCP immunoproteasome reveals a distinct mechanism of proteasome activation by PA28 $\alpha\beta$ by Chen et al. the authors provide structural models of the core proteasome bound to its activator PA28 $\alpha\beta$. By solving both the structure of the entire complex and the core proteasome they are able to reveal the structural adaptation of the core complex when bound to its activator. In this era of high-resolution cryoEM it is a nice piece of work, although from a technological point of view there are few advances to report. The binding of the proteasome activator PA28 $\alpha\beta$ is important for (MHC) class 1 antigen processing, especially for pathogen, and possibly tumor-derived, antigens. Reported are the cryo-electron microscopy (cryo-EM) structures of the free bovine core-proteasome, and of the single- and double-capped mammalian PA28 $\alpha\beta$ -iCP complexes and reasonable to good resolutions of around 3.5- 4.2 Å. As the mammalian core proteasome is built up from different α and β sub-units they use cross-linking mass spectrometry to find out which core proteasome subunits make most contact with the PA28 $\alpha\beta$ cap, also to address how the activation influence the opening for peptide/protein entry into the proteasome.

The complex was reconstituted from the parts and that may potentially have an influence on the XL-MS data. Are the authors sure that in these reconstituted samples there is not an access of either one of the components (e.g. PA28 $\alpha\beta$ or iCP) as then it is harder to assign cross-links to the complex or originating from the subunit.

When purified from spleen it is expected that a substantial part is the immunoproteasome, but also in spleen "normal" proteasome is also present. May it be that they only selected particles in their averaging that belong the immunoproteasome. A proteomics experiment, or a blot, will also be able to indicate how much normal proteasome is present in their preps.

My main concern is with the cross-linking data.

In the main text it is rather unclear how the XL-MS was performed. In the M&M they state that data-dependent tandem mass spectrometry (MS/MS) analysis data were processed with pLink software (Yang et al., 2012) and Proteome Discoverer 2.2 xlinkx.

Is there a rationale for why they use these two different search engines, did they give the same results. Additionally, they do cite the pLink software reference (Yang et al., 2012), then they should also cite the XlinkX software (Liu et al. Nature Communications volume 8, Article number: 15473 (2017))

My main concern is with the XL-MS data. I would like to see all cross-links detected (with appropriate FDR control) and also see intra cross-links mapped on the structure and cross-links between the subunits of the core-proteasome. Only if all these make sense it give credibility to the cross-links they do mention in the paper, where they zoom in (and mention only the cross-links between the PA28 $\alpha\beta$ and iCP subunits). There seem to be nice validation reports for the EM data, but this is lacking for the XL-MS data. This should really be supplied as well. Circular plots of all detected XL in both the iCP prep and the full complex should be presented, whereby I would not be surprised if some cross-links come from the "normal" CP particles.

Reviewer #2 (Remarks to the Author):

Paper from Chen et al. describes cryo-EM structures of mammalian PA28 $\alpha\beta$ -immunoproteasome (iCP), which is responsible for generating antigenic peptides. iCP works at the downstream of the 26S proteasome and it harbors three alternative proteolytic subunits: β 1i, β 2i, β 5i instead of constitutive

subunits $\beta 1$, $\beta 2$, and $\beta 5$. The iCP forms a complex not only with PA700 but also with a non-ATPase activator PA28 $\alpha\beta$ or PA28 γ and generates peptides that are more immunogenic than the ones generated by the constitutive proteasome (cCP). The authors purified the iCP from mice spleen where iCPs are abundantly expressed. They in vitro reconstituted the PA28 $\alpha\beta$ -iCP complex by crosslinking iCP with PA28 $\alpha\beta$ heptamer purified from bacteria. The structures showed detailed interaction between PA28 $\alpha\beta$ and iCP, which clarified a unique gate-opening mechanism of the iCP. They observed conformational change in iCP upon binding of the PA28 $\alpha\beta$. A structural comparison could explain the structural basis for the production of antigenic peptides in iCP. Although the structural basis for the CP gate opening by PA28 $\alpha\beta$ helped our understanding of the regulation of the mammalian CP gate, the many other aspects have been described in the proceeding papers. As the structures were solved in the presence of crosslinkers, the biological significance of the interface has to be addressed. Although fascinating from many viewpoints, most of the conclusions are lack of biological supports and it requires some additional experimental evidence to be accepted by Nature communications.

Major issues:

Immunoproteasome subunits $\beta 1i$, $\beta 2i$, $\beta 5i$ are IFN- γ -inducible proteasome subunits. In other words, constitutive proteolytic subunits, $\beta 1$, $\beta 2$, and $\beta 5$, are also expressed under normal conditions. It has been reported that tissues that express immunoproteasome such as liver, colon, intestine express both constitutive and immune proteolytic subunits simultaneously and 30-50% iCP contains constitutive subunits. That means the purified iCP may contain constitutive subunits. Although the authors showed some examples of side chain densities in Figure 1H, where the iCP subunits fit better than cCP. It is also possible that the side chain densities are averaged out by reconstruction. Is there any violation or ambiguous unassigned density that can't be explained by iCP? Besides, even though the authors showed the MS result in the S1B, they did not mention the other compositions, especially constitutive proteolytic subunits. The authors should show the MS results of full composition and quantify the occupancy of iCP subunits.

The interface between PA28 $\alpha\beta$ and iCP is unique and different from previous reports (pfPA28 and PA26). However, the interaction was stabilized by glutaraldehyde. It was not clear to me why mammalian PA28 $\alpha\beta$ is bound to iCP differently from pfPA28, even though the activation loops are highly conserved among them. In addition, it should be biologically addressed and commented on what is the determinant of the specific interaction between mammalian PA28 $\alpha\beta$ subunits and individual α -subunits. The activation loops are highly conserved between PA28 α and β while the C-terminal tails show variance. Does the activation loop contribute to the specific interaction? Figure 4 shows clusters of N-termini of α subunits in the CP gate. The structural basis for the gate opening is not well discussed. How do the insertions of the activation loops trigger the gate opening? Is the gate-opening mechanism conserved with the 26S proteasome? The authors should also prepare a figure which explains the more detailed structures.

Binding of PA28 $\alpha\beta$ induces shifts of iCP subunits such as $\alpha 3$, $\alpha 5$, $\beta 4$, and $\beta 6$. How does the conformational change alter the environment of the proteolytic sites? For example, the PA200-cCP structure from da Fonseca group reported the conformational change in the catalytic site upon binding of PA200.

Minor issues:

- Are three EM maps of iCP are equally resolved? How much uncertainty is there in modeling? Throughout all figures, it is not clear of which iCP densities and/or models were used to prepare the figures. They should describe it in the legend.
- In Figure S3A, it would be helpful to align PA26, pfPA28 sequences and indicate the position of the activation loops.

Reviewer #3 (Remarks to the Author):

The 20S proteasome core particle (20S CP) is a multi-catalytic large protease complex highly conserved in eukaryotes. Along with the standard 20S CP, which contains three catalytic subunits ($\beta 1$,

$\beta 2$, and $\beta 5$), it has been known that the immunoproteasome core particle (iCP) exists in mammalian cell, in which the catalytic subunits are replaced with the specific subunits, $\beta 1i$, $\beta 2i$, and $\beta 5i$. These CPs themselves are basically latent because of their closed gates, so there are various activators, such as the 19S particle, PA28 $\alpha\beta$, PA28 γ , and PA200 to open the gates of the CPs.

In this manuscript, the authors present the complete architectures of the mammalian PA28 $\alpha\beta$ -iCP immunoproteasome and free iCP by cryo-EM. This set of PA28 $\alpha\beta$ and iCP is especially important, because both complexes are induced by INF- γ , and are thought to be involved in the production of MHC class I-binding ligands. To date, the structures of the yeast 20S CP with TbPA26, which is a counterpart of PA28 in *Trypanosoma brucei*, and the PfPA28-20S CP complex from *Plasmodium falciparum* have been reported. Although all of these PA28 complexes bind to and activate the latent CPs by opening the CP gate, surprisingly, the data presented in this manuscript suggested that the binding mode and mechanism of association between the PA28 $\alpha\beta$ and the iCP is different from the other two sets. Thus, this reviewer thinks that the structures of the mammalian PA28 $\alpha\beta$ -iCP immunoproteasome presented for the first time in this manuscript are quite valuable and informative. This reviewer has only some concerns mainly about the presentation of the data listed below.

1. This reviewer does not understand the reason why the iCP seems more opened from the top view in Figure 1A than in Figure 1B, because one α -ring without PA28 $\alpha\beta$ should be closed in Figure 1A, whereas both α -rings are expected to be opened in Figure 1B.
2. It is hard to verify whether the conformational changes of the reverse turn regions lead to the gate opening of the iCP from the data in Fig. 3F as the authors claimed. the authors should present the data more clearly.
3. The picture is too small to appreciate the differences between occupied and partially-occupied pockets and between strong and weak c-ter-insertions in Fig. 4D.
4. It might be better to cite a paper by Santos R.L.A. et al (Nat. Commun. 2017 doi: 10.1038/s41467-017-01760-5.) for human iCP (PDB:6AVO).

Response to reviewer #1's comments:

Reviewer #1 (Remarks to the Author):

In this work entitled Cryo-EM of mammalian PA28 $\alpha\beta$ -iCP immunoproteasome reveals a distinct mechanism of proteasome activation by PA28 $\alpha\beta$ by Chen et al. the authors provide structural models of the core proteasome bound to its activator PA28 $\alpha\beta$. By solving both the structure of the entire complex and the core proteasome they are able to reveal the structural adaptation of the core complex when bound to its activator. In this era of high-resolution cryoEM it is a nice piece of work, although from a technological point of view there are few advances to report. The binding of the proteasome activator PA28 $\alpha\beta$ is important for (MHC) class 1 antigen processing, especially for pathogen, and possibly tumor-derived, antigens. Reported are the cryo-electron microscopy (cryo-EM) structures of the free bovine core-proteasome, and of the single- and double-capped mammalian PA28 $\alpha\beta$ -iCP complexes and reasonable to good resolutions of around 3.5- 4.2 Å. As the mammalian core proteasome is built up from different α and β sub-units they use cross-linking mass spectrometry to find out which core proteasome subunits make most contact with the PA28 $\alpha\beta$ cap, also to address how the activation influence the opening for peptide/protein entry into the proteasome.

Q1-1: The complex was reconstituted from the parts and that may potentially have an influence on the XL-MS data. Are the authors sure that in these reconstituted samples there is not an access of either one of the components (e.g. PA28 $\alpha\beta$ or iCP) as then it is harder to assign cross-links to the complex or originating from the subunit.

A1-1: Thanks for the insightful comment from the reviewer. Regarding the sample preparation for XL-MS analysis, we first mixed the purified and dialyzed PA28 $\alpha\beta$ and iCP at a molar ratio of 10:1 and incubated them at 37°C for 30 min to allow native associations between PA28 $\alpha\beta$ and iCP. We then cross-linked the PA28 $\alpha\beta$ -iCP complex by bis(sulfosuccinimidyl) suberate (BS3) with a final crosslinker concentration of 0.25 mM, and eventually used 20 mM Tris-HCl to terminate the reaction. Noteworthy, we performed further glycerol gradient centrifugation to remove un-crosslinked components in the reconstituted sample. Cross-linked PA28 $\alpha\beta$ -iCP complex was buffer exchanged to Hepes buffer, then subjected to XL-MS experiment. Furthermore, during XL-MS data analysis, we used E-value of $1.00E^{-02}$ as the threshold to remove extra lower-confidence XL-MS data corresponding to non-specific interactions, and only focused on the cross-links identified in the interface between PA28 $\alpha\beta$ and iCP units. This way to ensure we assigned the XLs between PA28 and iCP in their interface within the PA28 $\alpha\beta$ -iCP complex, instead of originating from the subunit or non-specific interactions. We have modified related text accordingly in our revised manuscript (paragraph one on P. 7, and paragraph two on P. 22). For more experimental details, please also refer to A1-3.

Q1-2: When purified from spleen it is expected that a substantial part is the immunoproteasome, but also in spleen "normal" proteasome is also present. May it be

that they only selected particles in their averaging that belong the immunoproteasome. A proteomics experiment, or a blot, will also be able to indicate how much normal proteasome is present in their preps.

A1-2: The reviewer is right that “When purified from spleen it is expected that a substantial part is the immunoproteasome, but also in spleen “normal” proteasome is also present”. Indeed, this has been supported by our mass spectrometry analysis of the free 20S proteasome of bovine spleen (Table R1), showing that the abundance of immune- β subunits, indicated by PSM (peptide-spectrum-matches) value, is obviously higher than that of standard- β subunits (β 1i vs β 1 >5.2, β 2i vs β 2 >3.4, β 5i vs β 5 >2.9). This is in line with a recent report showing that immunoproteasome is the most important proteasome subtype (more than 70% of the total 20S proteasome pool) in spleen (Menneteau et al., 2019) (Fig. R1, adopted from Menneteau et al., 2019). We have now modified related text (paragraph two on P.5), included Table R1 as Table S1, and referred the Menneteau et al., 2019 paper in our revised manuscript.

Besides, we should mention that during particle-picking and extraction from cryo-EM images at such low contrast, it is impossible to distinguish iCP from cCP particles. In reality, we picked and extracted all the particles exhibiting the general features of 20S/PA28-20S for subsequent data processing. Our further reconstruction eventually led to the final reconstruction showing the iCP features, indicating that iCP is the dominant species instead of cCP in bovine spleen.

Table R1 Mass spectrometry results of the full composition of the free 20S proteasome from bovine spleen

Bovine spleen 20S	Coverage	PSMs	Peptides	Unique peptides
PSMB9 / β 1i	76	377	12	12
PSMB10 / β 2i	42	252	8	8
PSMB8 / β 5i	60	215	14	14
PSMB6 / β 1	70	72	11	11
PSMB7 / β 2	64	74	15	15
PSMB5 / β 5	68	74	17	17
PSMB3 / β 3	53	242	14	14
PSMB2 / β 4	94	390	24	23
PSMB1 / β 6	73	304	19	19
PSMB4 / β 7	50	305	9	9
PSMA6 / α 1	74	497	20	20
PSMA2 / α 2	68	318	15	15
PSMA4 / α 3	77	419	18	18
PSMA7 / α 4	70	375	21	21
PSMA5 / α 5	66	355	15	15
PSMA1 / α 6	90	324	25	25
PSMA3 / α 7	44	181	14	14

Fig. R1 Stoichiometries of the six proteasome 20S subtypes in different Human tissue determined by the LC-SRM method (Menneteau et al., 2019).

Q1-3: My main concern is with the cross-linking data.

In the main text it is rather unclear how the XL-MS was performed. In the M&M they state that data-dependent tandem mass spectrometry (MS/MS) analysis data were processed with pLink software (Yang et al., 2012) and Proteome Discoverer 2.2 xlinkx. Is there a rationale for why they use these two different search engines, did they give the same results. Additionally, they do cite the pLink software reference (Yang et al., 2012), then they should also cite the XlinkX software (Liu et al. Nature Communications volume 8, Article number: 15473 (2017)).

A1-3: Thanks for pointing this out. We have confirmed with the coauthor who performed the XL-MS experiment that the cross-linked peptides were identified and evaluated using pLink2 software (Lu et al., 2015). The XL-MS experiment was performed long ago and we apologize for forgetting to update the XL-MS method. We have now updated the “Cross-linking/mass spectrometry analysis” session in the M&M (paragraph two on P. 22), which is also listed below for the convenience of the reviewer and editor.

We mixed the individually purified and dialyzed PA28αβ and iCP at a molar ratio of 10:1 and incubated them at 37°C for 30 min. We then cross-linked the associated PA28αβ-iCP complex by bis(sulfosuccinimidyl) suberate (BS3), with a final crosslinker concentration of 0.25 mM. After incubation on ice for 2 h, 20 mM Tris-HCl was used to terminate the reaction. To remove un-crosslinked components in the reconstituted sample, we performed further glycerol gradient centrifugation (15-45% glycerol (wt/vol), 25 mM Tris-HCl pH 7.4, 10 mM MgCl₂, 1 mM DTT) for 16 hrs at 37,000 rpm. We identified the fractions containing PA28αβ-iCP proteasomes by performing peptidase activity assay and NS-EM. Cross-linked complexes were buffer exchanged again to Hepes buffer with 1% glycerol and then subjected to MS experiment. The proteins were precipitated and digested for 16 hrs at 37°C by trypsin at an enzyme-to-substrate ratio of 1:50 (w/w). The tryptic digested peptides were desalted and loaded on an in-house packed capillary reverse-phase C18 column (40cm length, 100 μM ID x 360 μM

OD, 1.9 μM particle size, 120 \AA pore diameter) connected to an Easy LC 1200 system. The samples were analyzed with a 120 min-HPLC gradient from 6% to 35% of buffer B (buffer A: 0.1% formic acid in Water; buffer B: 0.1% formic acid in 80% acetonitrile) at 300 nL/min. The eluted peptides were ionized and directly introduced into a Q-Exactive mass spectrometer using a nano-spray source. Survey full-scan MS spectra (from m/z 300–1800) were acquired in the Orbitrap analyzer with resolution $r = 70,000$ at m/z 400. Cross-linked peptides were identified and evaluated using pLink2 software (Lu et al., 2015).

Q1-4: My main concern is with the XL-MS data. I would like to see all cross-links detected (with appropriate FDR control) and also see intra cross-links mapped on the structure and cross-links between the subunits of the core-proteasome. Only if all these make sense it give credibility to the cross-links they do mention in the paper, where they zoom in (and mention only the cross-links between the PA28 $\alpha\beta$ and iCP subunits). There seem to be nice validation reports for the EM data, but this is lacking for the XL-MS data. This should really be supplied as well. Circular plots of all detected XL in both the iCP prep and the full complex should be presented, whereby I would not be surprised if some cross-links come from the “normal” CP particles.

A1-4: The point is well taken. All cross-links were filtered by pLink2 with a separated FDR setting at 5% as default (Lu et al., 2015) and E-value was computed for further evaluation. To better visualize all cross-links detected by our XL-MS analysis, here we follow the suggestion from the reviewer to generate the circular plots through the xVis crosslink analysis server (<https://xvis.genzentrum.lmu.de/login.php>) (Fig. R2A). Here we only show the spectra data with E-value below 1.00E^{-02} in the circular plot. We also followed the suggestion to generate the plot with detected intra cross-links mapped on the core structure (Fig. R2C), which all fulfill the spatial geometry constrains of the linked amino acids (Liu and Heck, 2015; Yu et al., 2015). Furthermore, we take $\alpha 2$ and $\alpha 6$ core subunits as examples to illustrate the detected cross-links between the subunits of the core-proteasome, which shows that $\alpha 6$ forms XLs with $\alpha 5/\alpha 7$, while $\alpha 2$ forms XLs with $\alpha 1/\alpha 7$ (Fig. R2D). Mapping of these detected inter-subunit XLs on the core particle structure suggested that all those XLs meet the constrains of their relative spatial positions (Fig. R2E). These results validate the reliability of our XL-MS data.

As suggested by the reviewer, to further validate the XL-MS data, we carry out additional XL-MS analysis on the free iCP (Fig. R2F). Of note, since the sample and reagents have to be freshly prepared after two years from our original experiments, we tested the BS3 XL condition again and increased its concentration to 2mM, higher than that in our previous experiment (0.25 mM). Under this condition, also taking $\alpha 2$ and $\alpha 6$ subunits as examples, the detected XLs in the free iCP (Fig. R2F-G) overall matches the XLs detected in the PA28-iCP complex (Fig. R2B, D), with slightly increased number of detected XLs which might be related to the increased crosslinker concentration. This further validate the reliability of our XL-MS results.

Also, in both XL-MS experiments for PA28-iCP complex and free iCP, it appears there are only minor population of the detected XLs involving the constitutive β subunits ($\beta 1$,

$\beta 2$, and $\beta 5$), while the detected XLs involve more of the immune- β subunits ($\beta 1i$, $\beta 2i$, and $\beta 5i$) (Fig. R2B, F), consistent with our mass spectrometry result (Table R1) and the recent report (Menneteau et al., 2019).

Fig. R2 (A) Circular plot of all cross-links of PA28-iCP complex detected by XL-MS analysis, with intra-subunit XLs shown in red, and inter-subunit XLs in blue. We only show the spectra data with E-value below $1.00E^{-02}$ in the circular plot, which is followed throughout. (B) XLs of the iCP portion from the XL-MS results of the PA28-iCP complex. (C) Plot with detected intra subunit cross-links mapped on the core structure. (D) We

take $\alpha 2$ and $\alpha 6$ core subunits from PA28-iCP complex as examples to illustrate the detected XLs between the subunits of the core proteasome. (E) Mapping of those detected intra- and inter-subunit XLs for $\alpha 2$ and $\alpha 6$ subunits on the core particle structure. Due to the dynamic nature of the C-terminus, the $\alpha 6$ C-terminal residue K243 was not resolved in our structure, thus we could not display the XL of $\alpha 6(K243)-\alpha 7(K57)$ in the model; still the $\alpha 7(K57)$ (indicated by dotted black ellipsoid) is fairly close to $\alpha 6$ subunit. (F) Circular plot of detected XLs from the free iCP by XL-MS analysis. (G) Detected XLs for $\alpha 2$ (left) and $\alpha 6$ (right) from the free iCP XL-MS data.

Reviewer #2 (Remarks to the Author):

Paper from Chen et al. describes cryo-EM structures of mammalian PA28 $\alpha\beta$ -immunoproteasome (iCP), which is responsible for generating antigenic peptides. iCP works at the downstream of the 26S proteasome and it harbors three alternative proteolytic subunits: $\beta 1i$, $\beta 2i$, $\beta 5i$ instead of constitutive subunits $\beta 1$, $\beta 2$, and $\beta 5$. The iCP forms a complex not only with PA700 but also with a non-ATPase activator PA28 $\alpha\beta$ or PA28 γ and generates peptides that are more immunogenic than the ones generated by the constitutive proteasome (cCP). The authors purified the iCP from mice spleen where iCPs are abundantly expressed. They in vitro reconstituted the PA28 $\alpha\beta$ -iCP complex by crosslinking iCP with PA28 $\alpha\beta$ heptamer purified from bacteria. The structures showed detailed interaction between PA28 $\alpha\beta$ and iCP, which clarified a unique gate-opening mechanism of the iCP. They observed conformational change in iCP upon binding of the PA28 $\alpha\beta$. A structural comparison could explain the structural basis for the production of antigenic peptides in iCP. Although the structural basis for the CP gate opening by PA28 $\alpha\beta$ helped our understanding of the regulation of the mammalian CP gate, the many other aspects have been described in the proceeding papers. As the structures were solved in the presence of crosslinkers, the biological significance of the interface has to be addressed. Although fascinating from many viewpoints, most of the conclusions are lack of biological supports and it requires some additional experimental evidence to be accepted by Nature communications.

Major issues:

Q2-1: Immunoproteasome subunits $\beta 1i$, $\beta 2i$, $\beta 5i$ are IFN- γ -inducible proteasome subunits. In other words, constitutive proteolytic subunits, $\beta 1$, $\beta 2$, and $\beta 5$, are also expressed under normal conditions. It has been reported that tissues that express immunoproteasome such as liver, colon, intestine express both constitutive and immune proteolytic subunits simultaneously and 30-50% iCP contains constitutive subunits. That means the purified iCP may contain constitutive subunits. Although the authors showed some examples of side chain densities in Figure 1H, where the iCP subunits fit better than cCP. It is also possible that the side chain densities are averaged out by reconstruction. Is there any violation or ambiguous unassigned density that can't be explained by iCP?

A2-1: Thanks for the insightful comments from the reviewer. As suggested by our mass spectrometry analysis (Table R1), the immune- β subunits ($\beta 1i$, $\beta 2i$, and $\beta 5i$) are dominantly populated (> 70%) than the standard- β subunits ($\beta 1$, $\beta 2$, and $\beta 5$), suggesting that iCP is the main core particles in the purified bovine spleen proteasome, while cCP is rather less populated (please also refer to A1-2). This is in line with a recent report (Fig. R1) (Menneteau et al., 2019). Furthermore, we also showed that for those distinct regions between iCP and cCP, overall iCP model fits into our cryo-EM map better, a common strategy to distinguish iCP from cCP in previous studies (Huber et al., 2012). This further suggests that the dominantly populated iCP particles primarily contributed to our cryo-EM maps.

Still, since cCP also exists in the sample although the population is low, it may contribute faintly to the map. We carefully examined whether there are sidechain densities that cCP may fit in better than that of iCP. As has been shown in our Fig. 1H, most of the sidechain densities match the iCP model very well. There are rare locations as shown below in Fig. R3, when cCP $\beta 2$ S131 is fitted in, there appears to have extra density; while this extra density can be partially filled up by iCP Q131, implying the partial contribution of the S131 from cCP in this site in addition to that of Q131 from iCP. Still, Q131 is the better fit to this density possibly due to the dominate population of iCP. Collectively, rather than being averaged out, the dominantly populated iCP particles still primarily contribute to result in an iCP map; and our map resolution is high enough to unambiguously distinguish iCP from cCP for most of the specific sidechain densities. We have modified our description accordingly in the revised manuscript (last paragraph on P.6).

Fig. R3 Identification of immunoproteasome amino-acid residues by model and map fitting details. The zoomed in views in the second row show that for the same CP density, when cCP $\beta 2$ S131 is fitted in, there appears to have extra density (right panel); while this extra density can be partially filled up by iCP Q131 (left panel), implying the partial contribution of the S131 from cCP in this site in addition to that of Q131 from iCP. Still, Q131 of iCP is the better fit to this density possibly due to the dominate population of iCP. Also, in the first row, the Y25 in cCP (right panel) is completely out of the density.

Q2-2: Besides, even though the authors showed the MS result in the S1B, they did not mention the other compositions, especially constitutive proteolytic subunits. The authors should show the MS results of full composition and quantify the occupancy of iCP subunits.

A2-2: The point is well taken. We have now included the MS results of full composition as Table S1 in our revised manuscript, which suggests that the iCP is the dominant proteasome core particle in this sample (> 70%). We have modified related text in the

revised manuscript (paragraph two on P. 5). For more details, please also refer to A1-2 and Table R1 there.

Q2-3: The interface between PA28 $\alpha\beta$ and iCP is unique and different from previous reports (pfPA28 and PA26). However, the interaction was stabilized by glutaraldehyde. It was not clear to me why mammalian PA28 $\alpha\beta$ is bound to iCP differently from pfPA28, even though the activation loops are highly conserved among them.

A2-3: In recent years, crosslinking has become a more adopted strategy in cryo-EM studies to stabilize fragile micromolecular complexes (Ke et al., 2020; Lander et al., 2012; Poepsel et al., 2018; Yan et al., 2019), including the homologous pfPA28-20S complex (Xie et al., 2019). Considering the fragile nature of the PA28 $\alpha\beta$ -iCP complex (Dubiel et al., 1992), we added trace amount of crosslinker to stabilize the complex for it to sustain the purification and vitrification processes. Still, as has been pointed out that “fixation is not expected to give rise to any new conformations” (Ke et al., 2020).

Although the activation loops are highly conserved among PA28 $\alpha\beta$ and pfPA28, the activation loop–reverse turn interaction mainly contribute to destabilize the closed-gate conformation. Instead, the PA28 $\alpha\beta$ C-terminal tail insertions into the α -ring pockets of iCP provide the binding energy between PA28 $\alpha\beta$ and iCP (Whitby et al., 2000). Noteworthy, in PA28 $\alpha\beta$ -iCP both PA28 $\alpha\beta$ and iCP units are heteroheptameric, very distinct from that of pfPA28-20S with both pfPA28 and 20S being homoheptameric, and the sequence identity between PA28 α/β and pfPA28 is not high (~25%, Fig. S5). Therefore, the diversified properties in the key structural elements involving in the two units binding (including the activator C-terminal tails and the α -ring pockets of the 20S) could be the main reason why mammalian PA28 $\alpha\beta$ binds to the iCP differently from that of pfPA28.

Q2-4: In addition, it should be biologically addressed and commented on what is the determinant of the specific interaction between mammalian PA28 $\alpha\beta$ subunits and individual α -subunits. The activation loops are highly conserved between PA28 α and β while the C-terminal tails show variance. Does the activation loop contribute to the specific interaction?

A2-4: The point is well taken. It has been reported that in PA26/PAN-20S systems, the activator C-terminal tail insertions into the CP pockets can stabilize the binding, but is not sufficient to open the CP gate (Forster et al., 2005; Smith et al., 2007; Whitby et al., 2000). The subsequent interactions between the activation loops of activator and the reverse turns of the CP α -subunits in part lead to the CP gate opening (Forster et al., 2005; Forster et al., 2003; Whitby et al., 2000). Still, to further identify the key structural elements involving in the interaction and activation of iCP by PA28 $\alpha\beta$, we performed additional biochemical mutagenesis and deletion experiments on the recombinant human PA28 α/β subunits expressed in *E. coli*. Our binding analysis and proteolytic activity assay showed that after C-terminal truncations (truncated K245-Y249 for PA28 α and E234-Y239 for PA28 β , together termed PA28 $\alpha\beta^{\Delta C\text{-tails}}$), PA28 $\alpha\beta^{\Delta C\text{-tails}}$ cannot bind iCP (Fig. R4A-B), consequently, it failed to activate iCP (Fig. R4D).

These biochemical data substantiate the notion that the PA28 $\alpha\beta$ C-terminal tail insertions into the iCP pockets can stabilize the binding but is not sufficient to open the iCP gate, in line with previous reports of the homologous PA26-CP systems (Forster et al., 2005; Forster et al., 2003; Smith et al., 2007; Whitby et al., 2000).

Furthermore, our binding assay and the proteolytic activity assay together showed that the activation loop double mutated PA28 $\alpha^{N146Y}\beta^{N136Y}$ can still bind iCP (Fig. R4C), but failed to activate iCP (Fig. R4D); while the only PA28 β mutated PA28 $\alpha\beta^{N136Y}$ decreased the activation of iCP by about half (Fig. R4D), in agreement with previous biochemical studies on the PA28 activator (Huber and Groll, 2017; Zhang et al., 1998). Collectively, our data indicate that the activation loops especially N146 in PA28 α and N136 in PA28 β are the key structural elements in the activation and gate opening of iCP. Altogether, our structural and biochemical data suggest that the PA28 $\alpha\beta$ activation loops could involve in the specific interaction with the reverse turn of related α subunits of iCP, still it contributes more to the gate opening of iCP. We have now included Fig. R4 as Fig. S4 and updated related texts in the revised manuscript (first paragraph on P. 8, and first paragraph on P. 9).

Fig. R4 Binding assay (through NS-EM analysis) and proteolytic activity assay of wild type PA28 $\alpha\beta$ or mutated PA28 $\alpha\beta$ in complex with iCP. (A) Representative NS-EM image of iCP incubated with PA28 $\alpha\beta$. (B) Similar analyses as in (A) but for iCP incubated with the C-terminal truncated PA28 $\alpha\beta^{\Delta C-tails}$ (truncated K245-Y249 for PA28 α and E234-Y239 for PA28 β , together termed PA28 $\alpha\beta^{\Delta C-tails}$). (C) Similar analyses as in (A) but for iCP incubated with activation loop mutated PA28 $\alpha^{N146Y}\beta^{N136Y}$. (D) Proteolytic activity assay of iCP with PA28 $\alpha\beta$, PA28 $\alpha\beta^{\Delta C-tails}$, PA28 $\alpha^{N146Y}\beta^{N136Y}$, and PA28 $\alpha\beta^{N136Y}$ against the fluorogenic peptide Suc-LLVY-AMC. RFU, relative fluorescence units.

Q2-5: Figure 4 shows clusters of N-termini of α subunits in the CP gate. The structural basis for the gate opening is not well discussed. How do the insertions of the activation loops trigger the gate opening? Is the gate-opening mechanism conserved with the 26S proteasome? The authors should also prepare a figure which explains the more detailed structures.

A2-5: Thanks for the suggestion from the reviewer. In our manuscript, we focused more on the allosteric network coordination of the complex in the process of PA28 $\alpha\beta$ induced iCP gate opening. Our study suggested that the binding of PA28 $\alpha\beta$ with iCP would associate with the insertion of the C-terminal tails of four consecutive PA28 $\alpha\beta$ subunits (β_3 , α_3 , β_2 , and α_2) into the corresponding α -pockets ($\alpha_1/2$, $\alpha_2/3$, $\alpha_3/4$, and $\alpha_4/5$) of iCP, causing PA28 $\alpha\beta$ to lean towards the α_3 - α_4 side of iCP. The close proximity in this position could facilitate the formation of interactions between the activation loops of PA28 $\alpha\beta$ and the reverse turns of the iCP α -subunits, especially the $\alpha_2/3/4$ gate keepers, resulting in a shift in their reverse turn regions. Our further biological truncation and mutagenesis analyses suggested that the C-terminal tails (K245-Y249 and E234-Y239 in PA28 α and PA28 β , respectively) are crucial for PA28 $\alpha\beta$ binding iCP, and the activation loops especially N146 in PA28 α and N136 in PA28 β are the key structural elements for the gate opening of iCP (please refer to A2-4 and Fig. R4). Collectively, we conclude the interactions between the reverse turns of $\alpha_2/3/4$ and the corresponding PA28 $\alpha\beta$ activation loops (especially N146 in PA28 α and N136 in PA28 β) would disrupt the allosteric network of the gate keepers, resulting in a partially open gate of iCP. We have updated the related description (paragraph two on P.11).

For more detailed mechanism of activation loops induced gate opening, it has been well discussed in previous studies on PA26/PAN/BIm10 systems (Forster et al., 2005; Rabl et al., 2008; Sadre-Bazzaz et al., 2010; Whitby et al., 2000). Activator binding causes a rotation of the α -subunits in 20S CP, resulting in the radial and lateral movement of the Pro17 away from the pore. Then the conserved residues in α -subunits (Tyr8, Asp9, Pro17, and Tyr26) form a cluster to stabilize the open-gate conformation (Fig. R5, adopted from (Forster et al., 2005)). Since this mechanism has been investigated in homologous system in previous studies, while in our current study the resolution in the PA28 $\alpha\beta$ -iCP interaction interface is not high enough to identify all these details, we were not allowed to prepare a figure in such structural details.

Moreover, the gate-opening mechanism of PA28-CP proteasome is not conserved with the 26S proteasome, for the activation loops only exist in 11S activators. We already had a detailed discussion on the diversified signals for gate opening on P.14/15 in our original manuscript, to read "The C-terminal tails of several Rpt subunits from the hexameric ATPase ring in 26S inserted into the related α -ring pockets of CP, without direct interactions formed by activation loops and reverse turns as in the case of PA28 $\alpha\beta$. Although the activation seems to occur through different structural elements, they all regulate the allosteric network of the same enzymatic machine, i.e., the 20S CP, and eventually disturb the allosteric network in the key $\alpha_2/3/4$ gate keepers to

trigger the gate opening, readying the proteasome for substrate translocation and processing within the enzymatic core”.

Fig. R5 (A) The clusters of conserved residues (α Tyr8, α Asp9, α Pro17, and α Tyr26) that stabilize the open conformation are shown. (B) Enlarged view of cluster boxed in panel (A). *T. acidophilum* proteasome, blue; yeast proteasome, yellow. Both proteasome structures shown as seen in complex with PA26 after global alignment of the rings of α subunits. Adopted from (Forster et al., 2005).

Q2-6: Binding of PA28 $\alpha\beta$ induces shifts of iCP subunits such as α 3, α 5, β 4, and β 6. How does the conformational change alter the environment of the proteolytic sites? For example, the PA200-cCP structure from da Fonseca group reported the conformational change in the catalytic site upon binding of PA200.

A2-6: The point is well taken. We have performed further surface property analysis on the proteolytic sites of the immune-subunits. After association with PA28 $\alpha\beta$, the electrostatic surface of β 1i and β 2i proteolytic sites exhibit slightly weaker positive electrostatic potential compared with that of free iCP; in β 2i, a surrounding loop with negative charge properties appears moving inward towards the proteolytic site relative to the free iCP; while for β 5i, there isn't much obvious changes in surface property (Fig. R6). The subtle conformational changes and surface property variations induced by PA28 $\alpha\beta$ association may contribute to the regulation of iCP immunoproteasome activity, especially for β 1i and β 2i. We have now included Fig. R6 as Fig. S6B and added related description in the revised manuscript (first paragraph on P. 12).

Fig. R6 Variations of electrostatic surface property of the three catalytic subunits of bovine iCP induced by PA28 $\alpha\beta$ binding. Comparison of electrostatic surface property of β 1i, β 2i, and β 5i between free bovine iCP (top row) and PA28 $\alpha\beta$ -iCP (bottom row). The proteolytic site location is indicated by black dashed line. In β 2i, the surrounding negatively charged loop (indicated by yellow arrow head) shows inward movement towards the proteolytic site after binding the PA28 $\alpha\beta$.

Minor issues:

Q2-7: Are three EM maps of iCP are equally resolved? How much uncertainty is there in modeling?

A2-7: These cryo-EM maps were obtained through the same-level of data processing (Fig. S2A). Due to particle number differences and the free iCP has a C2 symmetry, the local resolution of free iCP is mostly better than 3.0 Å, and that of the iCP portion in PA28 $\alpha\beta$ -iCP and PA28 $\alpha\beta$ -iCP-PA28 $\alpha\beta$ maps is mostly around 3.5 Å (Fig. 1E-G). Still, at this resolution, for PA28 $\alpha\beta$ -iCP and PA28 $\alpha\beta$ -iCP-PA28 $\alpha\beta$, especially for their iCP portion, the models can be reliability generated and they fit into the map reasonably good (Fig. R7). We have now included Fig. R7 as Fig. S2C in our revised manuscript.

Fig. R7 High-resolution structure features and model-map fitting for PA28 $\alpha\beta$ -iCP and PA28 $\alpha\beta$ -iCP-PA28 $\alpha\beta$ complexes in the iCP portion.

Q2-8: Throughout all figures, it is not clear of which iCP densities and/or models were used to prepare the figures. They should describe it in the legend.

A2-8: Thanks for pointing this out. We have now added these details in the figure legends in our revised manuscript (figure legends of Fig. 1H and Fig. 3B-F).

Q2-9: In Figure S3A, it would be helpful to align PA26, pfPA28 sequences and indicate the position of the activation loops.

A2-9: The point is well taken. Instead of in Fig. S3A, where we intend to show for PA28 $\alpha\beta$ the sequence identify between human and mouse is high, we have now indicated the position of the activation loops by dotted red frames in Fig. S5, where we showed the sequence alignment among hPA28 $\alpha\beta$, PA26, and pfPA28.

Reviewer #3 (Remarks to the Author):

The 20S proteasome core particle (20S CP) is a multi-catalytic large protease complex highly conserved in eukaryotes. Along with the standard 20S CP, which contains three catalytic subunits ($\beta 1$, $\beta 2$, and $\beta 5$), it has been known that the immunoproteasome core particle (iCP) exists in mammalian cell, in which the catalytic subunits are replaced with the specific subunits, $\beta 1i$, $\beta 2i$, and $\beta 5i$. These CPs themselves are basically latent because of their closed gates, so there are various activators, such as the 19S particle, PA28 $\alpha\beta$, PA28 γ , and PA200 to open the gates of the CPs.

In this manuscript, the authors present the complete architectures of the mammalian PA28 $\alpha\beta$ -iCP immunoproteasome and free iCP by cryo-EM. This set of PA28 $\alpha\beta$ and iCP is especially important, because both complexes are induced by INF- γ , and are thought to be involved in the production of MHC class I-binding ligands. To date, the structures of the yeast 20S CP with TbPA26, which is a counterpart of PA28 in *Trypanosoma brucei*, and the PfPA28-20S CP complex from *Plasmodium falciparum* have been reported. Although all of these PA28 complexes bind to and activate the latent CPs by opening the CP gate, surprisingly, the data presented in this manuscript suggested that the binding mode and mechanism of association between the PA28 $\alpha\beta$ and the iCP is different from the other two sets. Thus, this reviewer thinks that the structures of the mammalian PA28 $\alpha\beta$ -iCP immunoproteasome presented for the first time in this manuscript are quite valuable and informative. This reviewer has only some concerns mainly about the presentation of the data listed below.

Q3-1. This reviewer does not understand the reason why the iCP seems more opened from the top view in Figure 1A than in Figure 1B, because one α -ring without PA28 $\alpha\beta$ should be closed in Figure 1A, whereas both α -rings are expected to be opened in Figure 1B.

A3-1: Thanks for pointing this out. In the top views in Fig. 1A and 1B, for better visualization of the PA28 $\alpha\beta$, we only see through PA28 $\alpha\beta$ to the top α -ring of iCP, instead of through the entire complex. To make it more clear, we added a small inset in Fig. 1A (also show it below) to illustrate the viewing angle and region where we generated the top views. The seemingly more open configuration in Fig. 1A is not the gate of iCP, but the extra pieces of density extending on the top of PA28 $\alpha\beta$ core, which most likely derived from the dynamic apical loops of PA28 $\alpha\beta$. They are too dynamic to be well resolved and appear discontinuous especially in the single capped complex, probably the apical loops are more dynamic in the single capped complex than in the double capped complex.

Fig. 1A Cryo-EM map of the single-capped PA28 $\alpha\beta$ -iCP complex (side view, enlarged top view and one central section). The inset illustrates the visualization angle and region, which was followed in the top view generation of PA28 $\alpha\beta$ -iCP-PA28 $\alpha\beta$.

Q3-2. It is hard to verify whether the conformational changes of the reverse turn regions lead to the gate opening of the iCP from the data in Fig. 3F as the authors claimed. the authors should present the data more clearly.

A3-2: The main purpose of Fig. 3F is to show the shifts of the reverse turns in α 2/3/4/5 subunits, which could result in repositions of the N-termini of α subunits. Similar phenomenon had been well documented in the PA26/PAN/Blm10 systems, suggesting that minor conformational changes in reverse turns induced by activator binding could disturb the allosteric network in the gate region, and lead to the gate opening of CP (Forster et al., 2005; Sadre-Bazzaz et al., 2010; Whitby et al., 2000). We have now added more descriptions in our revised manuscript (paragraph two on P. 9). For more details, please also refer to A2-5 and Fig. R5.

Q3-3. The picture is too small to appreciate the differences between occupied and partially-occupied pockets and between strong and weak c-ter-insertions in Fig. 4D.

A3-3: We have now followed the suggestion from the reviewer to enlarge Fig. 4D for better illustration. For the convenience of the reviewer and editor, we also show it below.

Fig. 4D A schematic diagram depicting a proposed mechanism of gate opening induced by PA28 $\alpha\beta$ binding.

Q3-4. It might be better to cite a paper by Santos R.L.A. et al (Nat. Commun. 2017 doi: 10.1038/s41467-017-01760-5.) for human iCP (PDB:6AVO).

A3-4: The point is well taken and we have now cited the literature for human iCP suggested by the reviewer in our revised manuscript (Nat. Commun. 2017 doi: 10.1038/s41467-017-01760-5) (please see first paragraph on P. 4).

References:

- Dubiel, W., Pratt, G., Ferrell, K., and Rechsteiner, M. (1992). Purification of an 11 S regulator of the multicatalytic protease. *The Journal of biological chemistry* 267, 22369-22377.
- Forster, A., Masters, E.I., Whitby, F.G., Robinson, H., and Hill, C.P. (2005). The 1.9 Å structure of a proteasome-11S activator complex and implications for proteasome-PAN/PA700 interactions. *Mol Cell* 18, 589-599.
- Forster, A., Whitby, F.G., and Hill, C.P. (2003). The pore of activated 20S proteasomes has an ordered 7-fold symmetric conformation. *EMBO J* 22, 4356-4364.
- Huber, E.M., Basler, M., Schwab, R., Heinemeyer, W., Kirk, C.J., Groettrup, M., and Groll, M. (2012). Immuno- and constitutive proteasome crystal structures reveal differences in substrate and inhibitor specificity. *Cell* 148, 727-738.
- Huber, E.M., and Groll, M. (2017). The Mammalian Proteasome Activator PA28 Forms an Asymmetric alpha4beta3 Complex. *Structure* 25, 1473-1480 e1473.
- Ke, Z., Oton, J., Qu, K., Cortese, M., Zila, V., McKeane, L., Nakane, T., Zivanov, J., Neufeldt, C.J., Cerikan, B., *et al.* (2020). Structures and distributions of SARS-CoV-2 spike proteins on intact virions. *Nature*.
- Lander, G.C., Estrin, E., Matyskiela, M.E., Bashore, C., Nogales, E., and Martin, A. (2012). Complete subunit architecture of the proteasome regulatory particle. *Nature* 482, 186-191.
- Liu, F., and Heck, A.J. (2015). Interrogating the architecture of protein assemblies and protein interaction networks by cross-linking mass spectrometry. *Curr Opin Struct Biol* 35, 100-108.
- Lu, S., Fan, S.B., Yang, B., Li, Y.X., Meng, J.M., Wu, L., Li, P., Zhang, K., Zhang, M.J., Fu, Y., *et al.* (2015). Mapping native disulfide bonds at a proteome scale. *Nature methods* 12, 329-331.
- Menneteau, T., Fabre, B., Garrigues, L., Stella, A., Zivkovic, D., Roux-Dalvai, F., Mouton-Barbosa, E., Beau, M., Renoud, M.L., Amalric, F., *et al.* (2019). Mass Spectrometry-based Absolute Quantification of 20S Proteasome Status for Controlled Ex-vivo Expansion of Human Adipose-derived Mesenchymal Stromal/Stem Cells. *Molecular & cellular proteomics : MCP* 18, 744-759.
- Poepsel, S., Kasinath, V., and Nogales, E. (2018). Cryo-EM structures of PRC2 simultaneously engaged with two functionally distinct nucleosomes. *Nature structural & molecular biology* 25, 154-162.
- Rabl, J., Smith, D.M., Yu, Y., Chang, S.C., Goldberg, A.L., and Cheng, Y. (2008). Mechanism of gate opening in the 20S proteasome by the proteasomal ATPases. *Molecular cell* 30, 360-368.
- Sadre-Bazzaz, K., Whitby, F.G., Robinson, H., Formosa, T., and Hill, C.P. (2010). Structure of a Blm10 complex reveals common mechanisms for proteasome binding and gate opening. *Molecular cell* 37, 728-735.
- Smith, D.M., Chang, S.C., Park, S., Finley, D., Cheng, Y., and Goldberg, A.L. (2007). Docking of the proteasomal ATPases' carboxyl termini in the 20S proteasome's alpha ring opens the gate for substrate entry. *Molecular cell* 27, 731-744.
- Whitby, F.G., Masters, E.I., Kramer, L., Knowlton, J.R., Yao, Y., Wang, C.C., and Hill, C.P. (2000). Structural basis for the activation of 20S proteasomes by 11S regulators. *Nature* 408, 115-120.
- Xie, S.C., Metcalfe, R.D., Hanssen, E., Yang, T., Gillett, D.L., Leis, A.P., Morton, C.J., Kuiper, M.J., Parker, M.W., Spillman, N.J., *et al.* (2019). The structure of the PA28-20S proteasome complex from *Plasmodium falciparum* and implications for proteostasis. *Nat Microbiol* 4, 1990-2000.
- Yan, K., Yang, J., Zhang, Z., McLaughlin, S.H., Chang, L., Fasci, D., Ehrenhofer-Murray, A.E., Heck, A.J.R., and Barford, D. (2019). Structure of the inner kinetochore CCAN complex assembled onto a centromeric nucleosome. *Nature* 574, 278-282.
- Yu, C., Mao, H., Novitsky, E.J., Tang, X., Rychnovsky, S.D., Zheng, N., and Huang, L. (2015). Gln40 deamidation blocks structural reconfiguration and activation of SCF ubiquitin ligase complex by Nedd8. *Nat Commun* 6, 10053.
- Zhang, Z.G., Clawson, A., Realini, C., Jensen, C.C., Knowlton, J.R., Hill, C.P., and Rechsteiner, M. (1998). Identification of an activation region in the proteasome activator REG alpha. *P Natl Acad Sci USA* 95, 2807-2811.

REVIEWERS' COMMENTS

Reviewer #1 (Remarks to the Author):

The authors have done sufficiently to overcome my initial criticism on the XL-MS data and better described the work and data. I find it however a pity that they describe this only in the rebuttal letter. I would advocate to promote the figure in the rebuttal (Fig. R2) at least to the supplementary data of the article, and also copy some of the relevant text into this

Reviewer #2 (Remarks to the Author):

Chen et al have sincerely addressed many concerns and suggestions from the three Reviewers to improve their paper. Their MS data indicate that approximately 30% of particles are constitutive proteasome. Although the iCP is dominant, 30% is a significant number, especially considering that the resolution is around 3.0 -3.5 Å, in which the small side chains are not distinguishable. I agree that their structure can be used to understand the interaction between iCP and PA28qβ because the interface with PA28qβ does not contain the immunoproteasome subunits (β1i, β2i, β5i are located at the inner-rings). I also understand their argument that with a low S/N ratio it is not possible to differentiate the cCP and iCP by classification. I am simply concerned whether it is scientifically correct to call their EM map as iCP structure when 30% of signals are originated from the constitutive proteasome. In the manuscript, they described as if the cCP signal can be negligible but it is fair to describe that their structural information contains 30% of the cCP signal. I would also recommend showing Fig. R3 in Supplementary. It would be nice the author discussed their structure more precisely in terms of the contamination of the cCP.

PMS is not precise to argue about the quantification. There are several methods established to quantify proteins with label-free approaches (e.g. Cox et al., MCP 2014). It would be great if the authors can provide protein quantification.

It is interesting to see that the PA28qβN136Y complex which harbors a single mutation in PA28βN136Y decreased the activation of iCP by about half. How about the single mutation in PA28α?

Reviewer #3 (Remarks to the Author):

The authors have adequately addressed the comments. I therefore now support publication.

REVIEWERS' COMMENTS

Reviewer #1:

Q1-1: The authors have done sufficiently to overcome my initial criticism on the XL-MS data and better described the work and data. I find it however a pity that they describe this only in the rebuttal letter. I would advocate to promote the figure in the rebuttal (Fig. R2) at least to the supplementary data of the article, and also copy some of the relevant text into this.

A1-1: Thanks for the constructive suggestion from the reviewer. We have promoted Fig. R2 into the supplementary data as Supplementary Fig. 4 and modified related text accordingly in our revised manuscript (paragraph one on P. 7). To read, “The cross-links detected within the CP of proteasome fulfill the spatial geometry constrains of the linked amino acids, validating the reliability of our XL-MS data (Supplementary Fig. 4)”.

Reviewer #2:

Q2-1: Chen et al have sincerely addressed many concerns and suggestions from the three Reviewers to improve their paper. Their MS data indicate that approximately 30% of particles are constitutive proteasome. Although the iCP is dominant, 30% is a significant number, especially considering that the resolution is around 3.0 -3.5 Å, in which the small side chains are not distinguishable. I agree that their structure can be used to understand the interaction between iCP and PA28 $\alpha\beta$ because the interface with PA28 $\alpha\beta$ does not contain the immunoproteasome subunits (β 1i, β 2i, β 5i are located at the inner-rings). I also understand their argument that with a low S/N ratio it is not possible to differentiate the cCP and iCP by classification. I am simply concerned whether it is scientifically correct to call their EM map as iCP structure when 30% of signals are originated from the constitutive proteasome. In the manuscript, they described as if the cCP signal can be negligible but it is fair to describe that their structural information contains 30% of the cCP signal. I would also recommend showing Fig. R3 in Supplementary. It would be nice the author discussed their structure more precisely in terms of the contamination of the cCP.

A2-1: Thanks for the positive comments on our responding works. The point is well taken, we have now included Fig. R3 as Supplementary Fig. 2d in our revised

manuscript. As shown in Supplementary Table 1, the abundance of immune- β subunits β 1i, β 2i, and β 5i is 83.9%, 77.3%, and 74.4%, respectively. As shown in A2-2 and Table R1, the iBAQ indicated abundance of immune- β subunits β 1i, β 2i, and β 5i is 69.7%, 89.4%, and 84.6%, respectively, also obviously higher than that of standard- β subunits. Taken together, in general 70% is a relative conservative estimation of the iCP population in the purified bovine spleen proteasome.

Moreover, for the free iCP, dominant portion of the map showed local resolution levels better than 3.0 Å (Fig. 1g), revealing side chain densities for most of the amino acids. Careful comparison of the distinct regions between iCP and cCP suggested that most of the sidechain densities match the iCP model very well and overall iCP model fits into our cryo-EM map better (Fig. 1h), and only rare locations implied the partial contribution of the signal from cCP in addition to the dominant signal from iCP (Supplementary Fig. 2d). So judging from the map itself, it exhibits representative iCP characteristics. This is a common strategy to distinguish iCP from cCP in previous studies¹. We therefore call the EM map as iCP structure.

Still, we followed the suggestion to add a discussion to describe the contribution of cCP signal in the map formation in our revised manuscript, to read “..... for instance, for these distinct regions between iCP and cCP, overall iCP model fits into our cryo-EM map better (Fig. 1h), **only rare locations imply the partial contribution from the signal of cCP in addition to the dominant signal from iCP (Supplemental Fig. 2d).** This further implied that the dominantly populated iCP particles (more than 70%) primarily contributed to our cryo-EM maps; **still, we cannot exclude the minor contribution from the signal of cCP.**” (paragraph two on P. 6).

Q2-2: PMS is not precise to argue about the quantification. There are several methods established to quantify proteins with label-free approaches (e.g. Cox et al., MCP 2014). It would be great if the authors can provide protein quantification.

A2-2: Thanks for the suggestion from our reviewer. We have reprocessed the dataset with the suggested method² using MaxQuant computational proteomics platform. As shown in Table R1, the abundance of immune- β subunits, indicated by iBAQ (intensity-based absolute-protein-quantification) value, is obviously higher than that of standard- β subunits, with the population abundance of β 1i, β 2i, and β 5i at 69.7%, 89.4%, and

84.6%, respectively. This is in line with the conclusion in our manuscript that the abundance of immune- β subunits is higher (>70%) than that of standard- β subunits.

Additionally, it has been described in previous literatures that “The peptide spectrum match (PSM) value represents the relative abundance of the target protein, with a higher PSM value indicating a higher abundance”³, suggesting that PSM is also an acceptable and common way to illustrate the relative abundance of the protein components³⁻⁵. Therefore, we prefer to keep Supplementary Table 1 showing the PSM data.

Table R1. Mass spectrometry analysis of the full composition of the free 20S proteasome from bovine spleen.

Bovine spleen 20S	Coverage	iBAQ	Peptides	Unique peptides
PSMB9 / β 1i	53	2.74E+09	12	12
PSMB10 / β 2i	42.5	6.61E+09	8	8
PSMB8 / β 5i	61.2	3.66E+09	14	5
PSMB6 / β 1	73.6	1.19E+09	12	12
PSMB7 / β 2	58.1	7.78E+08	13	13
PSMB5 / β 5	65.8	6.62E+08	17	17
PSMB3 / β 3	60.5	8.31E+09	16	16
PSMB2 / β 4	93.5	5.54E+09	24	24
PSMB1 / β 6	70.1	5.95E+09	16	16
PSMB4 / β 7	50.4	5.92E+09	9	9
PSMA6 / α 1	82.1	5.85E+09	26	26
PSMA2 / α 2	87.2	5.91E+09	20	20
PSMA4 / α 3	78.2	6.41E+09	20	20
PSMA7 / α 4	69.8	7.93E+09	22	22
PSMA5 / α 5	75.5	4.83E+09	18	18
PSMA1 / α 6	92.4	1.03E+10	29	29
PSMA3 / α 7	60.8	6.41E+09	16	16

Q2-3: It is interesting to see that the PA28 α β N136Y complex which harbors a single mutation in PA28 β N136Y decreased the activation of iCP by about half. How about the single mutation in PA28 α ?

A2-3: It has been shown in previous biochemical study that the N146 of PA28 α and the N136 of PA28 β play important role in proteasome activation⁶. We were inspired by their results and designed the activation loop mutations of N146Y in PA28 α and N136Y in PA28 β by two-step cloning: we first performed the N136Y mutation in PA28 β to generate PA28 α β ^{N136Y}, then further mutated it to PA28 α ^{N146Y} β ^{N136Y}. Thus,

we didn't generate the PA28 $\alpha^{N146Y}\beta$ mutation. Still, combining our proteolytic activity assay that the activation loop double mutated PA28 $\alpha^{N146Y}\beta^{N136Y}$ can still bind iCP, but failed to activate iCP; while the only PA28 β mutated PA28 $\alpha\beta^{N136Y}$ decreased the activation of iCP by about half (Supplementary Fig. 5d, also shown below), it is more likely that PA28 $\alpha^{N146Y}\beta$ could decrease the activity of iCP by about half.

Supplementary Figure 5d. Proteolytic activity assay of wild type PA28 $\alpha\beta$ or mutated PA28 $\alpha\beta$ in complex with iCP. The activation loop double mutated PA28 $\alpha^{N146Y}\beta^{N136Y}$ failed to activate iCP; while the only PA28 β mutated PA28 $\alpha\beta^{N136Y}$ decreased the activation of iCP by about half.

Reviewer #3:

The authors have adequately addressed the comments. I therefore now support publication.

Response: We thank the reviewer for the positive comments on our work.

References:

- 1 Huber, E. M. *et al.* Immuno- and constitutive proteasome crystal structures reveal differences in substrate and inhibitor specificity. *Cell* **148**, 727-738, doi:10.1016/j.cell.2011.12.030 (2012).
- 2 Cox, J. *et al.* Accurate proteome-wide label-free quantification by delayed normalization and maximal peptide ratio extraction, termed MaxLFQ. *Mol Cell Proteomics* **13**, 2513-2526, doi:10.1074/mcp.M113.031591 (2014).
- 3 Yan, C. *et al.* Structure of a yeast spliceosome at 3.6-angstrom resolution. *Science* **349**, 1182-1191, doi:10.1126/science.aac7629 (2015).
- 4 Wan, R. *et al.* The 3.8 Å structure of the U4/U6.U5 tri-snRNP: Insights into spliceosome assembly and catalysis. *Science* **351**, 466-475, doi:10.1126/science.aad6466 (2016).
- 5 Zhu, Y. *et al.* DEqMS: A Method for Accurate Variance Estimation in Differential Protein Expression Analysis. *Mol Cell Proteomics* **19**, 1047-1057, doi:10.1074/mcp.TIR119.001646 (2020).
- 6 Zhang, Z. G. *et al.* Identification of an activation region in the proteasome activator REG alpha. *P Natl Acad Sci USA* **95**, 2807-2811, doi:DOI 10.1073/pnas.95.6.2807 (1998).